# Inactivation of *Invs/Nphp2* in renal epithelial cells drives infantile nephronophthisis like phenotypes in mouse

**Yuanyuan Li[1†], Wenyan Xu[1†], Svetlana Makova[2], Martina Brueckner[2], Zhaoxia Sun[1*]**

[1]Department of Genetics, Yale University School of Medicine, New Haven, United States; [2]Department of Pediatrics, Yale University School of Medicine, New Haven, United States

**\*For correspondence:**
zhaoxia.sun@yale.edu

[†]These authors contributed equally to this work

**Competing interest:** The authors declare that no competing interests exist.

**Abstarct:** Nephronophthisis (NPHP) is a ciliopathy characterized by renal fibrosis and cyst formation, and accounts for a significant portion of end stage renal disease in children and young adults. Currently, no targeted therapy is available for this disease. *INVS/NPHP2* is one of the over 25 NPHP genes identified to date. In mouse, global knockout of *Invs* leads to renal fibrosis and cysts. However, the precise contribution of different cell types and the relationship between epithelial cysts and interstitial fibrosis remains undefined. Here, we generated and characterized cell-type-specific knockout mouse models of *Invs*, investigated the impact of removing cilia genetically on phenotype severity in *Invs* mutants and evaluated the impact of the histone deacetylase inhibitor valproic acid (VPA) on *Invs* mutants. Epithelial-specific knockout of *Invs* in *Invs*[flox/flox]*;Cdh16-Cre* mutant mice resulted in renal cyst formation and severe stromal fibrosis, while *Invs*[flox/flox]*;Foxd1-Cre* mice, where *Invs* is deleted in stromal cells, displayed no observable phenotypes up to the young adult stage, highlighting a significant role of epithelial-stromal crosstalk. Further, increased cell proliferation and myofibroblast activation occurred early during disease progression and preceded detectable cyst formation in the *Invs*[flox/flox]*;Cdh16-Cre* kidney. Moreover, concomitant removal of cilia partially suppressed the phenotypes of the *Invs*[flox/flox]*;Cdh16-Cre* mutant kidney, supporting a significant interaction of cilia and *Invs* function in vivo. Finally, VPA reduced cyst burden, decreased cell proliferation and ameliorated kidney function decline in *Invs* mutant mice. Our results reveal the critical role of renal epithelial cilia in NPHP and suggest the possibility of repurposing VPA for NPHP treatment.

## Editor's evaluation

Germline inactivation of NPHP2, which encodes a protein that localizes to the transition zone at the base of the primary cilium, results in infantile kidney cysts and fibrosis. In this study, the authors provide solid evidence that increased cell proliferation and fibrosis precede cyst formation in Nphp-2 mouse models, that mutant renal epithelial cells are responsible for the phenotype, and that genetic inhibition of ciliogenesis in this model reduces disease severity. They also show that valproic acid, a drug that affects a number of cellular targets and is used to treat other human conditions, slows disease progression.

## Introduction

Primary cilia are widely distributed cell surface organelles that function as signaling hubs for almost all vertebrate cell types (*Schneider et al., 2005*; *Rohatgi et al., 2007*; *Lancaster et al., 2011*; *Huangfu*

**eLife digest** One of the most common causes of kidney failure in children and young adults is nephronophthisis. This genetic disease causes cysts and tissue scarring in the kidneys, leading to excessive urine production and extreme tiredness. Unfortunately, there is no targeted therapy available for this condition.

Scientists do not fully understand how genetic mutations lead to these symptoms. Previous research in mice showed that blocking the gene for a protein called INVS recreated signs similar to nephronophthisis. However, it is not clear how the different cell types in the kidneys are involved. Previous results suggest that cilia, the hair-like projections on the surface of cells, could be involved in developing cysts in nephronophthisis.

To understand how the disease is driven, Li, Xu et al. created a range of genetically modified mice with INVS missing in different cell types. When INVS was removed from cells that line the kidney tubules, the mice developed scarring and cysts. By contrast, there were no symptoms when connective tissue cells were lacking INVS. When Li, Xu et al. removed the cilia from the cells, it helped to reduce the negative impact of the loss of INVS. In addition, a drug called valproic acid reduced the cysts and tissue scarring, and slowed kidney decline in the mutant mice, suggesting the possibility of repurposing this drug for nephronophthisis treatment.

These results could help researchers to study other conditions that are influenced by the health of cilia. Future work on nephronophthisis will be needed to understand how INVS causes the disease and the mechanism for the benefits of valproic acid.

*et al., 2003*; *Haycraft et al., 2005*; *Corbit et al., 2008*). Ciliary dysfunction is associated with an array of human disorders, collectively termed ciliopathies, that include obesity, mental retardation, retinal degeneration, and polycystic kidney disease (PKD), among many others (reviewed in *Hildebrandt et al., 2009*). In renal epithelial cells, cilia protrude into the lumen of tubules, exposed to both mechanical and chemical signals carried by extracellular fluid flow. Disruption of cilia biogenesis almost inevitably leads to the formation of epithelium-lined fluid filled kidney cysts in mouse (*Pazour et al., 2000*; *Yoder et al., 2002*).

Polycystin 1 and 2 (PC1 and PC2), encoded by the autosomal dominant PKD (ADPKD) genes *Pkd1* and *Pkd2*, respectively, are targeted to cilia and this specific trafficking highly correlates with their in vivo function (*Yoder et al., 2002*; *Barr and Sternberg, 1999*; *Cai et al., 2014*; *Pazour et al., 2002*; *Yoshiba et al., 2012*). Interestingly, although both cilia biogenesis and polycystin mutants develop cystic kidney, concomitant removal of cilia ameliorates cyst progression triggered by polycystin inactivation, demonstrating that intact cilia are required for cyst growth in ADPKD models (*Ma et al., 2013*). Based on these results, it was postulated that polycystins function to inhibit a novel and uncharacterized cilia-dependent cyst activating (CDCA) signaling pathway (*Ma et al., 2013*). This hypothesis implies the existence of a separate cyst-inhibiting (CDCI) pathway mediated by cilia, inactivation of which, together with CDCA, underlies the milder renal cyst formation in cilia biogenesis mutants compared to polycystin mutants. Renal fibrosis is also frequently observed in cystic kidney diseases; but is often considered as secondary to cyst formation and the underlying mechanism remains poorly understood.

Nephronophthisis (NPHP) is a recessive renal cystic disease with cysts mostly confined in the corticomedullary junction region in the juvenile form of the disease, and more widely distributed across the kidney in the infantile form. Since multiple NPHP proteins are localized to cilia, basal bodies and/or show functional involvement in cilia biogenesis and/or signaling, NPHP is considered as a ciliopathy (*Fliegauf et al., 2006*; *Otto et al., 2003*; *Otto et al., 2008*; *Shiba et al., 2010*; *Shiba et al., 2009*; *Williams et al., 2010*; *Winkelbauer et al., 2005*; *Won et al., 2011*). Compared with other types of renal ciliopathies, a prominent feature of NPHP is the severity of interstitial fibrosis. Genetically NPHP is highly heterogenous and at least 25 genes have been identified for the classic form of this disease (for a review, see *Srivastava et al., 2017*). However, mouse mutants of NPHP genes do not always show obvious renal phenotypes (*Won et al., 2011*; *Jiang et al., 2008*; *Ronquillo et al., 2016*). *Invs*, the homolog of which is associated with both infantile and juvenile NPHP in human, was discovered as a novel gene inactivated in a mouse mutant with reversed internal organ asymmetry along the

left-right body axis and was originally named as *Inversin* (*Invs*) (*Yokoyama et al., 1993*; *Mochizuki et al., 1998*; *Morgan et al., 1998*). This whole-body knockout model additionally displays kidney cysts in both the medulla and cortex and renal fibrosis at the neonatal stage, resembling the renal phenotypes of infantile NPHP (*Morgan et al., 1998*; *Phillips et al., 2004*). However, the functional significance of *Invs* in different renal cell types is undefined. Moreover, the progression of renal fibrosis and its relationship with cyst formation remains to be characterized.

Here, to dissect tissue-specific function of *Invs* and the mechanism of interstitial fibrosis resulting from defective *Invs*, we generated two conditional mouse knockout models: (1) targeting *Invs* deletion in epithelial cells of the distal nephron (*Invs^{flox/flox}*;*Cdh16-Cre* mice), and (2) deleting *Invs* in stromal cells (*Invs^{flox/flox}*;*Foxd1-Cre* mice). We found that the infantile NPHP-like phenotypes in mutant kidneys arise from abnormal signaling specifically within and from renal epithelial cells. Further, we investigated the relationship between cyst formation and interstitial fibrosis. Our results suggest that the fibrotic response and abnormal increase of cell proliferation precede and therefore could initiate independently of cyst formation in this model. Mechanistically, we find that removal of cilia reduces the severity of *Invs* phenotypes, suggesting that, similar to polycystins, INVS inhibits a cilia-dependent cyst and fibrosis activating pathway. Finally, we show that the histone deacetylase inhibitor (HDACI) valproic acid (VPA) reduces cell proliferation, suppresses cyst expansion, and preserves kidney function in *Invs* mutant mice, suggesting VPA, and perhaps other HDACIs, as a potential candidate drug for treating NPHP.

## Results

### Inactivation of *Invs* in renal epithelial cells of the distal nephron leads to progressive cystic kidney disease in mouse

To dissect tissue-specific function of *Invs*, we obtained embryonic stem cell clones generated by the European Conditional Mouse Mutagenesis Program (EUCOMM). The *Invs^{tm1a}* 'knockout-first' allele contains an IRES:lacZ-neo selection cassette located within two FLPE recombinase (FRT) sites and two loxP sites flanking the third exon of mouse *Invs* (*Figure 1A*). Deletion between the two loxP sites is predicted to lead to frame shift and a premature stop codon. The *Invs^{tm1a/+}* mice were crossed to a deleter strain expressing FLPe under the control of a ubiquitous actin promoter (*Rodríguez et al., 2000*) to generate a floxed allele of *Invs*. The carrier *Invs^{flox/+}* mice were backcrossed with C57BL/6 J for five generations and then crossed with carriers of *Cdh16-Cre,* which directs Cre expression in the distal nephron of the developing metanephros after E11.5 (*Shao et al., 2002a*; *Shao et al., 2002b*), to generate *Invs^{flox/+}*;*Cdh16-Cre* mice. Crossing *Invs^{flox/flox}* with *Invs^{flox/+}*;*Cdh16-Cre* mice led to the generation of live *Invs^{flox/flox}*;*Cdh16-Cre* pups in Mendelian ratio. In mutants, neonatal pups showed progressive enlargement of the kidney: At postnatal day 7 (P7), the mutant kidney size was comparable to that in control animals and there was no significant difference in body weight and kidney to body weight (KBW) ratio between the two groups (*Figure 1B and C*). At P14, the mutant kidneys were larger, and KBW ratios were significantly increased, although the body weight remained comparable (*Figure 1B and C*). Mutant kidneys became increasingly enlarged at P21 and P28, with comparable body weight at P21 and reduced body weight in mutants at P28 (*Figure 1B and C*). We additionally compared KBW ratios between male and female mice and no significant differences were detected between genders in either control or mutant groups at P14 and P21 (*Figure 1D*).

To inspect for potential cyst formation, we performed Hematoxylin/eosin (HE) stainning on kidney sections and quantified cystic index as defined previously (*Cao et al., 2009*; *Shibazaki et al., 2008*). Consistent with the progression of the KBW ratio, no cyst was detected in the P7 mutant kidney and the cystic index remained comparable between the wild type and mutant kidney (*Figure 1E and F*). At P14, small cysts were detected, and the cystic index was already significantly increased in the mutant kidney (*Figure 1E and F*). At P21, cysts became larger and more numerous in mutant kidneys (*Figure 1E*), and by P28 mutant kidneys were severely cystic with large cysts throughout the cortex and medullar region (*Figure 1E*). The cystic index progressively increased in mutant kidneys during the same time window (*Figure 1F*).

To understand the cellular origin of kidney cysts in mutant mice, we performed immunofluorescence staining, labeling different segments of the nephron in kidney sections. Consistent with histology, neither the collecting duct, labelled with Dolichos Biflorus Agglutinin (DBA), nor the proximal tubule,

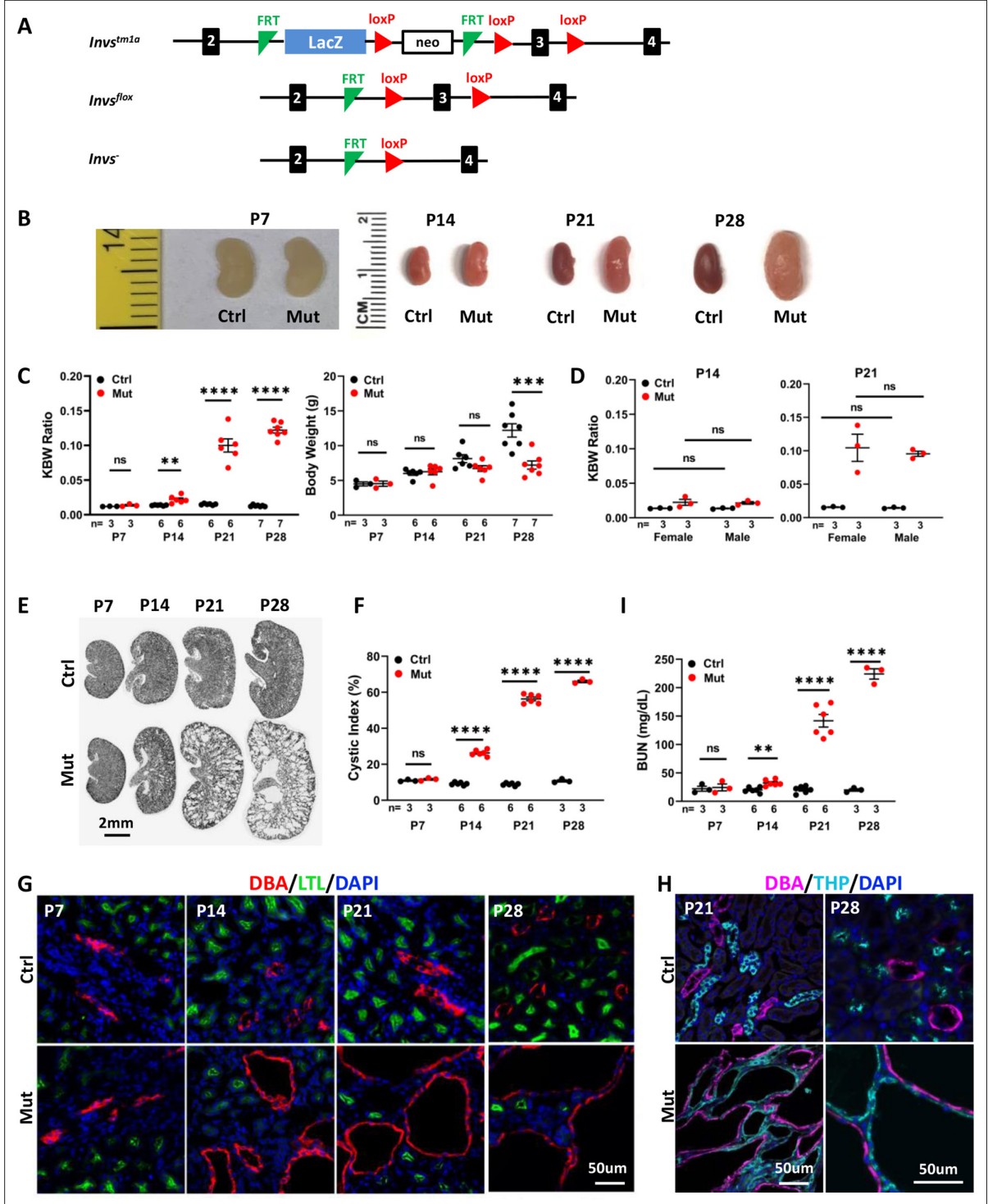

**Figure 1.** *Invs^{flox/flox};Cdh16-Cre* mice develop progressive cystic kidney disease at the neonatal stage. (**A**) Generation of a floxed allele of *Invs*. FRT indicates FLPE recombinase sites, and loxP indicates Cre recombinase sites. (**B**) Gross morphology of the kidney. (**C**) KBW ratio and body weight in control and mutant mice. (**D**) KBW ratio in male and female mice. (**E**) HE-stained kidney sections. (**F**) Cystic index. (**G, H**) Origin of cysts. DBA marks the collecting duct in red in G and in purple in H. The proximal tubule is labeled with LTL in green and the medullary thick ascending limb is labeled by THP in cyan. Nuclei are labeled with DAPI in blue. (**I**) BUN level. Ctrl: littermate control; Mut: *Invs^{flox/flox};Cdh16-Cre* mutants. Unpaired t-test was performed between mutants and control animals. Data are presented as mean ± SEM: ns: not significant (p>0.05); **: p<0.01; ***: p<0.001; ****: p<0.0001.

The online version of this article includes the following source data for figure 1:

**Source data 1.** Related to *Figure 1C, D, F and I*.

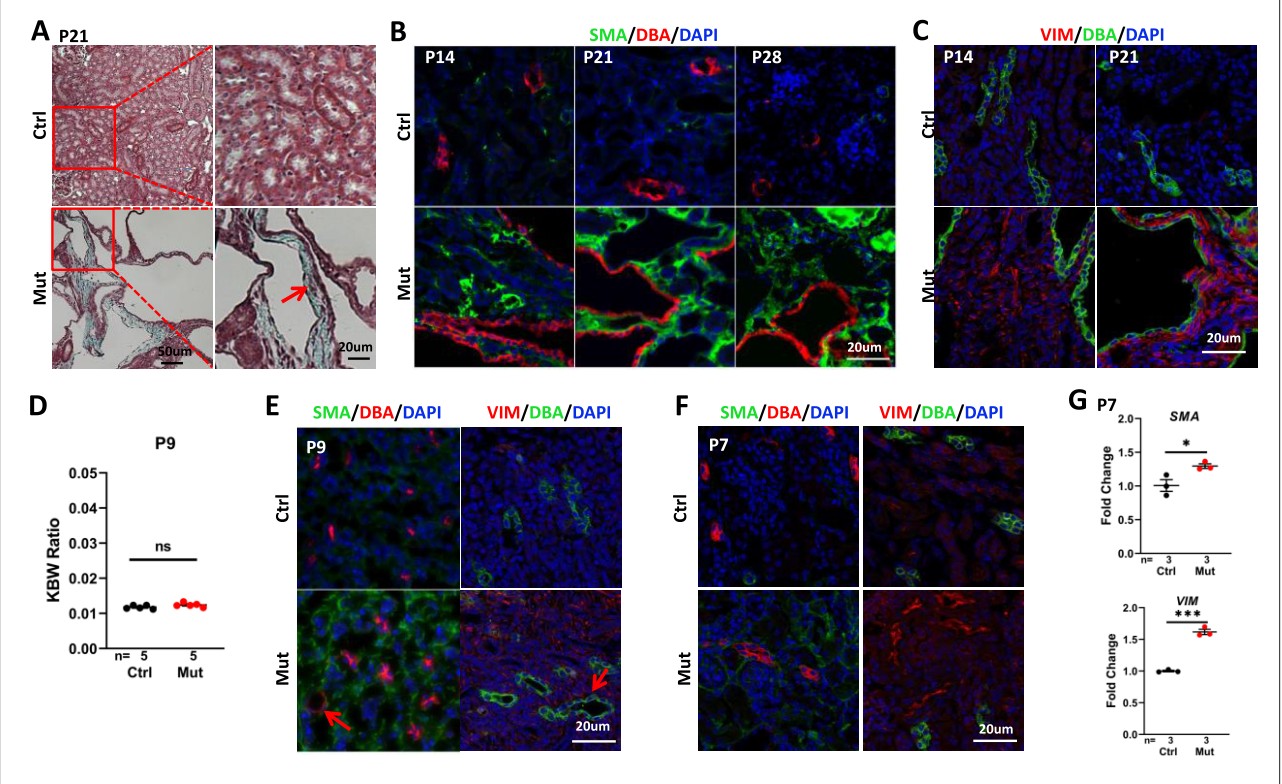

**Figure 2.** *Invs*<sup>flox/flox</sup>;*Cdh16-Cre* mice develop interstitial fibrosis in the kidney. (**A**) Trichome staining of P21 kidney sections indicates blue collagen deposition (arrow) in the cortex region of a mutant kidney. (**B**) Increased signal of SMA (green) in the cortex region of mutant kidneys from P14 to P28. DBA in red marks the collecting duct. DAPI in blue labels the nucleus. (**C**) Increase of vimentin staining (VIM, red) in the cortex region of P14 and P21 mutant kidneys. DBA in green. DAPI in blue. (**D**) KBW ratio at P9. (**E, F**) Modest increase of SMA (in green, DBA in red in the left panels) and vimentin (VIM in red in the right panels, DBA in green) in the cortex region of the mutant kidney at P9 (**E**) and P7 (**F**). Arrows point to dilated tubules. DAPI in blue labels the nucleus. (**G**) The level of *SMA* (upper) and *Vimentin* (lower) mRNA is increased in the mutant kidney as assayed by RT-qPCR using P7 whole kidney lysates. Ctrl: littermate control; Mut: *Invs*<sup>flox/flox</sup>;*Cdh16-Cre* mutants. Unpaired t-test was performed between mutants and control animals. Data are presented as mean ± SEM. ns: not significant (p>0.05); *: p<0.05; ***: p<0.001.

The online version of this article includes the following source data for figure 2:

**Source data 1.** Related to *Figure 2D and G*.

labelled with Lotus Tetragonolobus Lectin (LTL), showed enlargement at P7 (***Figure 1G***). At P14, cysts were detected mostly in the DBA positive collecting duct (***Figure 1G***). At P21 and P28, DBA positive cysts were enlarged and numerous (***Figure 1G***). Large cysts were also detected in the thick ascending limb, labelled by anti-Tamm-Horsfall protein (THP) at both P21 and P28 (***Figure 1H***). By contrast, the LTL positive proximal tubule was free from cysts at P28 (***Figure 1G***). Overall, the cystic region was consistent with the expression domain of *Cdh16-Cre*.

To monitor kidney function, we assayed for the blood urea nitrogen (BUN) level. At P7, mutant and control mice were comparable (***Figure 1I***). At P14, BUN levels became significantly higher in mutant mice (***Figure 1I***), and BUN levels became more elevated in mutant mice at P21 and P28, demonstrating a progressive decline of kidney function (***Figure 1I***).

Together, these results reveal a progressive renal cystic disease in *Invs*<sup>flox/flox</sup>;*Cdh16-Cre* mice.

### *Invs*<sup>flox/flox</sup>;*Cdh16-Cre* mice develop severe renal interstitial fibrosis

As interstitial fibrosis is a significant phenotype in NPHP, we first used trichrome staining to detect collagen deposition in P21 kidney sections. In mutant kidneys, an increase of the blue staining of collagen was detected in the cortex region (***Figure 2A***). Further, we performed immunostaining of smooth muscle actin (SMA), a marker of myofibroblast, to monitor the fibrotic response over time. In the wildtype kidney, SMA signal was confined to the peri-vascular region. SMA signal was increased in the cortex region of the mutant kidney at P14 and became dramatically elevated and widespread

in the mutant kidney at P21 and P28 (*Figure 2B*). We additionally performed immunostaining of vimentin, a marker of activated fibroblasts that becomes upregulated early during injury responses in the kidney (*Ó hAinmhire et al., 2019*; *Zhou et al., 2019*). Vimentin signal was increased in the cortex region of the mutant kidney at P14 and P21 (*Figure 2C*).

## Fibrotic response precedes cyst formation in *Invs^{flox/flox}*;*Cdh16-Cre* mice

To further delineate the relationship between cyst formation and interstitial fibrosis, we analyzed mutant kidney at earlier stages. At P9, the KBW ratio was normal in mutants (*Figure 2D*). By contrast, SMA and vimentin signal in the cortex region was already increased (*Figure 2E*). However, we also detected isolated tubule dilation on mutant kidney sections (*Figure 2E*). At the non-cystic P7 stage, SMA and vimentin signals were modestly increased in the cortex region (*Figure 2F*). Reverse transcription and quantitative PCR (RT-qPCR) using kidney lysates also showed a significant increase of *Sma* and *Vimentin* mRNA level in the mutant at P7 (*Figure 2G*).

Combined, these results reveal an increase of fibrotic markers at P7, preceding detectable cyst formation.

## Cell over-proliferation precedes cyst formation in the *Invs^{flox/flox}*;*Cdh16-Cre* kidney

Since over-proliferation of epithelial cells is a driver of cyst progression (*Nadasdy et al., 1995*; *Takakura et al., 2009*; *Zhang et al., 2021*), we performed immunofluorescence staining of P7 kidney sections using anti-proliferating cell nuclear antigen (PCNA), a nuclear marker for cell turnover. PCNA positive cells increased in the collecting duct (DBA+) in the cortex region of the mutant kidney (*Figure 3A and B*, upper panels). PCNA positive cells also increased outside of DBA positive regions. To investigate whether the number of proliferating interstitial cells were increased, we outlined epithelial tubules by co-labeling with anti-laminin. The number of proliferating interstitial cells, outside of epithelial regions encircled by laminin, was significantly increased in the mutant kidney (*Figure 3A and B*, lower panels). Taking advantage of the distinct localization patterns of PCNA at different cell cycle stages, that is replication foci in S-phase and diffusive nuclear localization in G1 and G2 cells (*Zerjatke et al., 2017*), we further showed that the number of cells in interphase and S phase, respectively, was significantly increased in both the collecting duct and interstitium (*Figure 3A and B*). Moreover, the number of mitotic cells, labelled by the mitosis marker phosphorylated histone H3 (pHH3), was significantly increased in both regions (*Figure 3C and D*).

Combined, these results revealed increased proliferating cells. through both cell autonomous and non-autonomous mechanisms, in pre-cystic *Invs^{flox/flox}*;*Cdh16-Cre* kidneys.

## *Invs^{flox/flox}*;*Foxd1-Cre* mice show no observable renal phenotypes up to the young adult stage

As interstitial fibrosis is a prominent phenotype of NPHP, we investigated the role of *Invs* in the stroma by generating and characterizing *Invs^{flox/flox}*;*Foxd1-Cre* mice. *Foxd1-Cre* drives Cre expression in metanephric mesenchymal cells destined to be stromal cells, including interstitial cells, mesangial cells and pericytes; and Cre activity can be readily detected at E11.5, similar to the onset of Cdh16-Cre activity (*Shao et al., 2002a*; *Shao et al., 2002b*; *Hum et al., 2014*; *Humphreys et al., 2010*). The resultant mutant mice were viable. At P28, we did not observe any morphologic or functional phenotypes. No differences were seen in kidney size, KWB ratio, cyst formation, or collagen deposition in the mutant kidney at P28 (*Figure 4A–E*). In addition, immunofluorescence staining of kidney sections with the connecting tubule and collecting duct marker aquaporin-2 (AQP2), the collecting duct maker DBA, the thick ascending limb marker THP and the proximal tubule marker LTL showed no difference in the mutant kidney (*Figure 4F*). Moreover, immunostaining of SMA failed to detect any changes in the mutant kidney at P28 (*Figure 4G*). In accordance, BUN levels were also normal in P28 mutants (*Figure 4H*). To verify deletion of *Invs* in interstitial cells, we first analyzed the expression of eGFP-Cre by immunostaining of eGFP and detected wide expression of eGFP-Cre in the renal stroma in both the cortex and medulla region (*Figure 4I*). We then cultured primary interstitial cells following a previously published protocol (*Nakai et al., 2021*) and performed immunostaining on cultured cells. We detected INVS in the proximal compartment of 85.01% of cilia in cells from control kidneys, consistent with the previously reported localization pattern of INVS (*Shiba et al., 2009*; *Bennett et al., 2020*), whereas

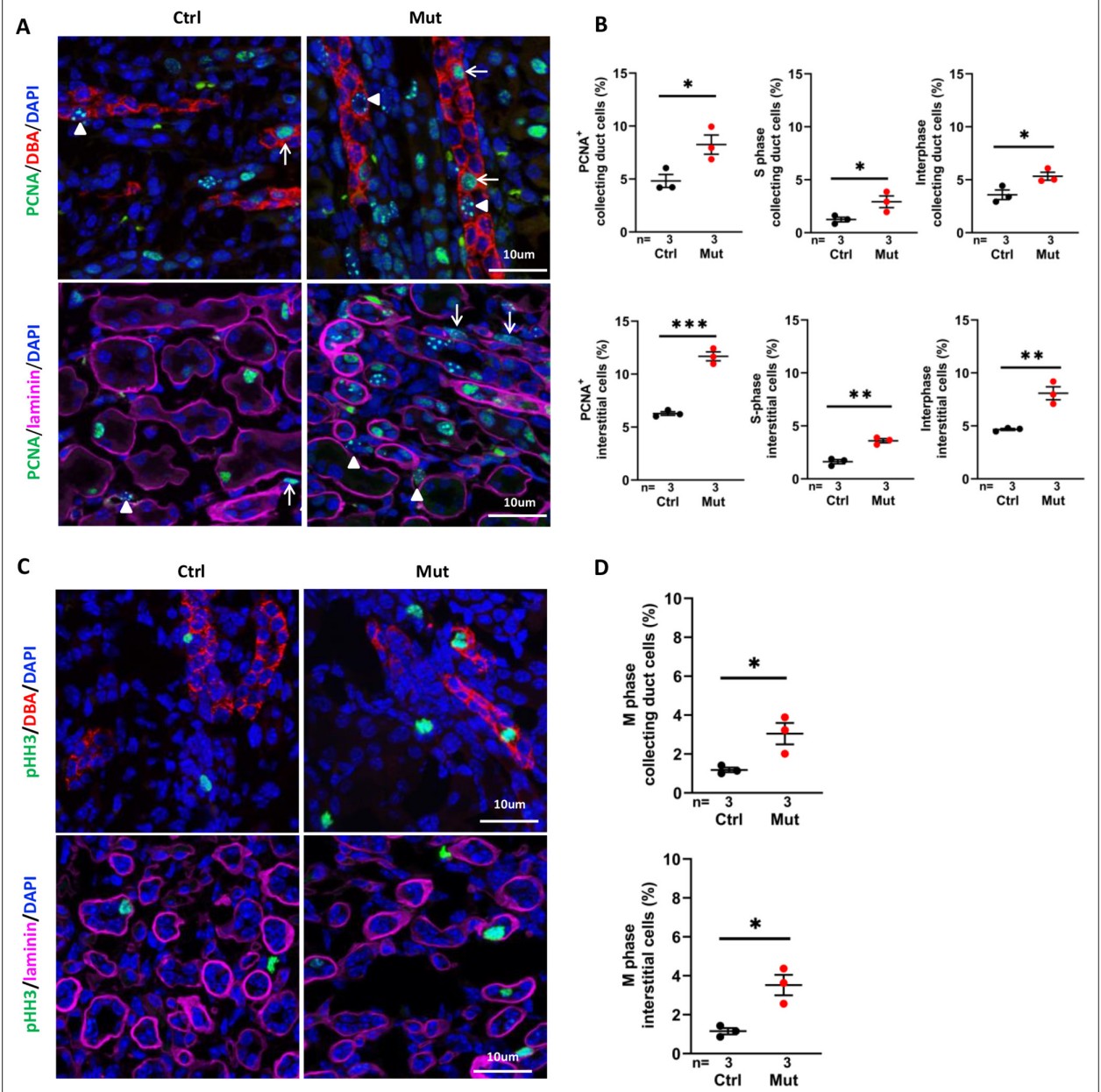

**Figure 3.** *Invs^flox/flox;Cdh16-Cre* kidneys show increased cell proliferation at P7. (**A**) Proliferating cells in the cortex region. PCNA (in green) stains replication foci in the nucleus of S-phase cells (examples pointed by arrowheads) but shows diffusive nuclear signal in interphase cells (arrows). In upper panels, collecting duct cells are labeled with DBA in red. In lower panels, epithelial regions are encircled by anti-laminin staining in purple. DAPI in blue labels the nucleus. (**B**) Percentage of total PCNA+ cells, PCNA+ S phase cells and PCNA+ interphase cells in collecting duct cells and interstitial cells in the cortex region. (**C**) Mitotic cells labelled by anti-pHH3 in the cortex region. In upper panels, collecting duct cells are labelled with DBA in red. In lower panels, epithelial regions are encircled by anti-laminin staining in purple. (**D**) Percentage of mitotic cells in collecting duct cells (upper panel) and interstitial cells (lower panel) in the cortex region. DAPI in blue labels the nucleus. Ctrl: littermate control; Mut: *Invs^flox/flox;Cdh16-Cre* mutants. Unpaired t-test was performed between mutants and control animals. Data are presented as mean ± SEM. *: p<0.05; **: p<0.01; ***: p<0.001.

The online version of this article includes the following source data for figure 3:

**Source data 1.** Related to *Figure 3B and D*.

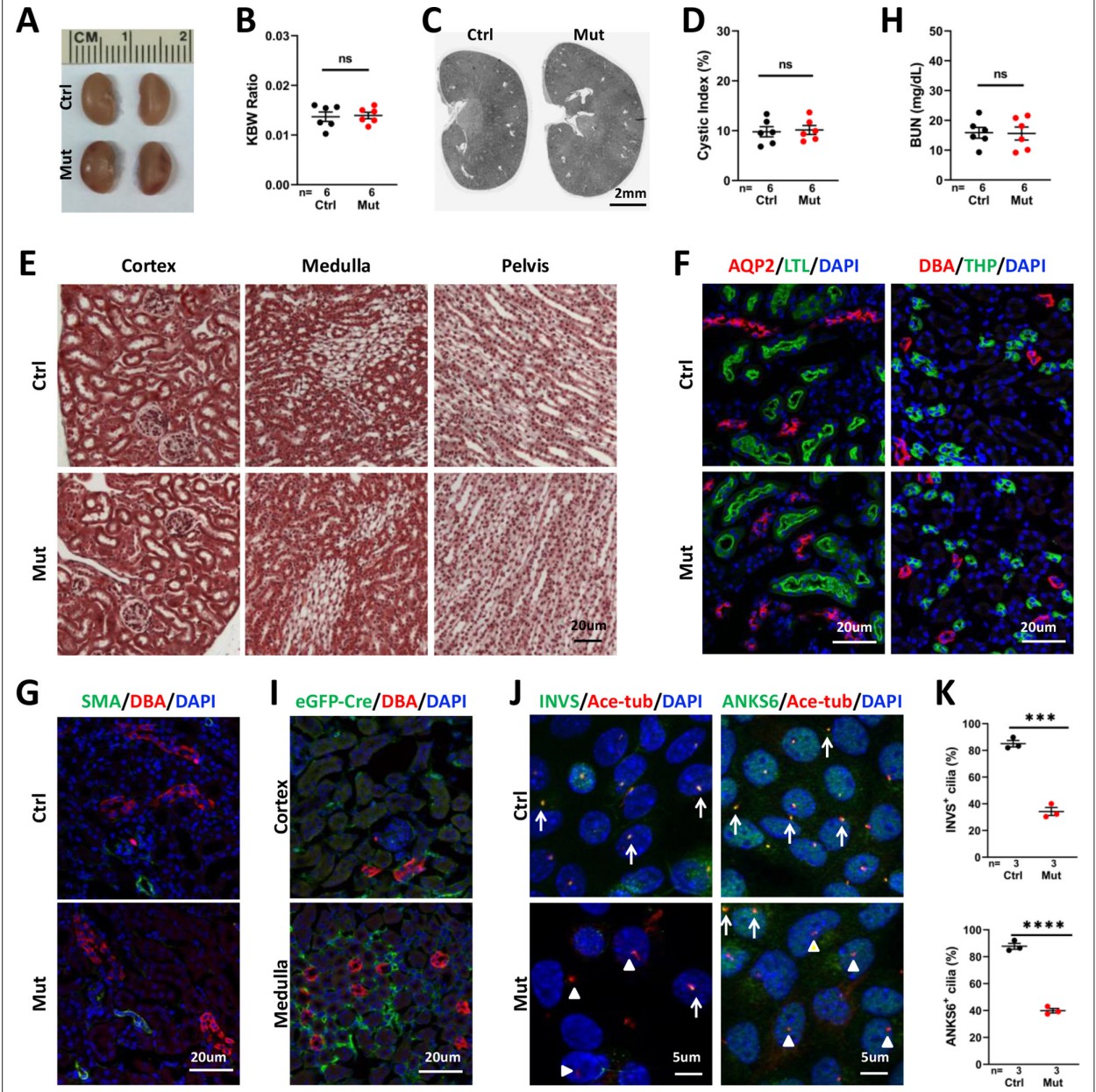

**Figure 4.** *Invs^{flox/flox}*;*Foxd1-Cre* kidneys show no significant phenotype at P28. (**A**) Gross morphology of the kidney. (**B**) KBW ratio. (**C**) HE-stained sections of the kidney. (**D**) Cystic index. (**E**) Trichrome staining of kidney sections. (**F**) Normal morphology of the collecting tubule and duct labelled by AQP2 (in red, left panels), collecting duct labelled by DBA (in red in right panels), proximal tubule (LTL in green in left panels) and thick ascending limb (THP in green in right panels). DAPI in blue labels the nucleus. (**G**) No increase of SMA signal (in green) can be detected in the mutant kidney. DBA in red labels the collecting duct. DAPI in blue labels the nucleus. (**H**) BUN level. (**I**) Immunostaining of eGFP-Cre (green). DBA in red. DAPI in blue. (**J**) Immunostaining of INVS (in green, left panels) and ANKS6 (in green, right panels) in primary culture of interstitial cells from control and mutant kidneys. Anti-acetylated tubulin (Ace-tub, red) labels cilia. DAPI in blue. Arrows indicate INVS or ANKS6 positive cilia. Arrowheads point to INVS⁻ or ANKS6⁻ cilia. (**K**) Quantification of INVS and ANKS6 positive cilia. Ctrl: littermate control; Mut: *Invs^{flox/flox}*;*Foxd1-Cre* mutants. Unpaired t-test was performed between mutants and control animals. Data are presented as mean ± SEM. ns: not significant (p>0.05); ***: p<0.001; ****: p<0.0001.

The online version of this article includes the following source data and figure supplement(s) for figure 4:

**Source data 1.** Related to *Figure 4B, D, H and K*.

**Figure supplement 1.** *Invs^{flox/flox}*;*Foxd1-Cre* kidneys show no significant phenotype at P56.

**Figure supplement 1—source data 1.** Related to *Figure 4—figure supplement 1A*.

only 34.24% INVS-positive cilia were observed in cells of mutant kidneys (*Figure 4J and K*). Similarly, ANKS6, encoded by another NPHP gene and is known to localize to the Inversin compartment in a INVS dependent manner (*Bennett et al., 2020*; *Czarnecki et al., 2015*), was detected in 87.78% of cilia in cells from the control kidney, but was reduced to 39.89% cilia in cells from the mutant kidney (*Figure 4J and K*), validating the deletion of *Invs* in a significant fraction of interstitial cells.

We then expanded our analysis to week 8. At P56, *Invs^flox/flox^;Foxd1-Cre* mice showed normal body weight, KBW ratio, renal histology and BUN, suggesting normal kidney structure and function (*Figure 4—figure supplement 1*).

Taken together, these results suggest that deletion of *Invs* in stromal cells fails to trigger renal fibrosis nor kidney cysts up to young adults.

## Genetic abrogation of cilia partially suppresses disease progression in *Invs^flox/flox^;Cdh16-Cre* mice

INVS is localized to and defines the Inversin compartment, a proximal segment of cilia (*Shiba et al., 2009*; *Bennett et al., 2020*). However, it is also detected in subcellular locations outside of the cilium (*Nürnberger et al., 2004*; *Nürnberger et al., 2002*; *Estrada Mallarino et al., 2020*). Moreover, INVS is dispensable for cilia biogenesis (*Phillips et al., 2004*; *Sang et al., 2011*). We validated this result in *Invs^flox/flox^;Cdh16-Cre* mice. At P21, cilia were present in cyst-lining DBA positive cells as indicated by immunostaining with the cilia marker anti-acetylated tubulin in the *Invs^flox/flox^;Cdh16-Cre* kidney (*Figure 5A*). In addition, ARL13B localization to the cilium was also undisrupted (*Figure 5B*).

The CDCA hypothesis posits that ectopic activation of a cilia-dependent cyst activating pathway underlies cyst formation in ADPKD (*Ma et al., 2013*). If INVS functions in this pathway, removal of cilia would extinguish the CDCA pathway and consequently rescue the phenotypes of *Invs* mutants. To test this hypothesis, we generated *Invs^flox/flox^;Ift88^flox/flox^;Cdh16-Cre* double knockout mice. *Ift88* is an intraflagellar transport (IFT) gene essential for cilia biogenesis (*Pazour et al., 2000*). Immunostaining with the cilia marker anti-acetylated tubulin confirmed abrogation of cilia in the distal nephron in both *Ift88^flox/flox^;Cdh16-Cre* single knockout and *Invs^flox/flox^;Ift88^flox/flox^;Cdh16-Cre* double knockout mice, as expected (*Figure 5C*). Consistent with previous reports, *Ift88^flox/flox^;Cdh16-Cre* single knockout mice showed a slower progression of renal cystic disease in comparison to *Invs^flox/flox^;Cdh16-Cre* mutants. At P21, neither kidney size nor KBW ratio were significantly changed in *Ift88^flox/flox^;Cdh16-Cre* mutants compared to age matched control mice (*Figure 5D and E*). In histological sections, *Ift88^flox/flox^;Cdh16-Cre* mutant kidneys contained only small cysts but a significant increase of cystic index (*Figure 5F and G*). By contrast, stage-matched *Invs^flox/flox^;Cdh16-Cre* single mutants had enlarged and severely cystic kidneys, with significantly increased KBW ratios and cystic index (*Figure 5D–G*). Interestingly, the increase of the KBW ratio and cystic index was reduced in *Invs^flox/flox^;Ift88^flox/flox^;Cdh16-Cre* double mutant mice in comparison to *Invs^flox/flox^;Cdh16-Cre* single mutant mice regardless of gender (*Figure 5E–G*). Moreover, the BUN level was also significantly lower in *Invs^flox/flox^;Ift88^flox/flox^;Cdh16-Cre* double mutants than in *Invs^flox/flox^;Cdh16-Cre* single mutants regardless of gender, suggesting a partial suppression of kidney function decline (*Figure 5H*).

We then investigated the impact of *Ift88* deletion on interstitial phenotypes observed in P21 *Invs^flox/flox^;Cdh16-Cre* mutant kidneys. Immunostaining of kidney sections revealed that the increase of of SMA and vimentin was more moderately in *Ift88^flox/flox^;Cdh16-Cre* mutants compared to *Invs^flox/flox^;Cdh16-Cre* mutants (*Figure 5I*). In addition, *Invs^flox/flox^;Ift88^flox/flox^;Cdh16-Cre* double mutants showed reduced signal of SMA and vimentin compared to *Invs^flox/flox^;Cdh16-Cre* single mutants (*Figure 5I*). Western Blot of total kidney lysates was then used to quantify the levels of SMA and Collagen I. While the levels of both proteins were significantly increased in *Invs* single mutants compared to controls, the increase in *Ift88* single mutants was significant by t-test but not by one-way ANOVA analysis followed by Šidák's test (*Figure 5J*). Moreover, the levels of SMA and Collagen I in double mutants were significantly reduced compared to *Invs* single mutants (*Figure 5J*). Combined, these results suggest that both myofibroblast activation and fibrosis were reduced when *Ift88* was deleted in *Invs^flox/flox^;Cdh16-Cre* mutants.

We further investigated the impact of *Ift88* deletion on the abnormal increase of cell proliferation in the cortex region of the *Invs^flox/flox^;Cdh16-Cre* mutant kidney. At P21, 0.9% of DBA positive collecting duct cells were PCNA positive in the control kidney, which was increased to 6.3% in the *Ift88^flox/flox^;Cdh16-Cre* mutant kidney and 22.9% in the *Invs^flox/flox^;Cdh16-Cre* mutant kidney (*Figure 6A*

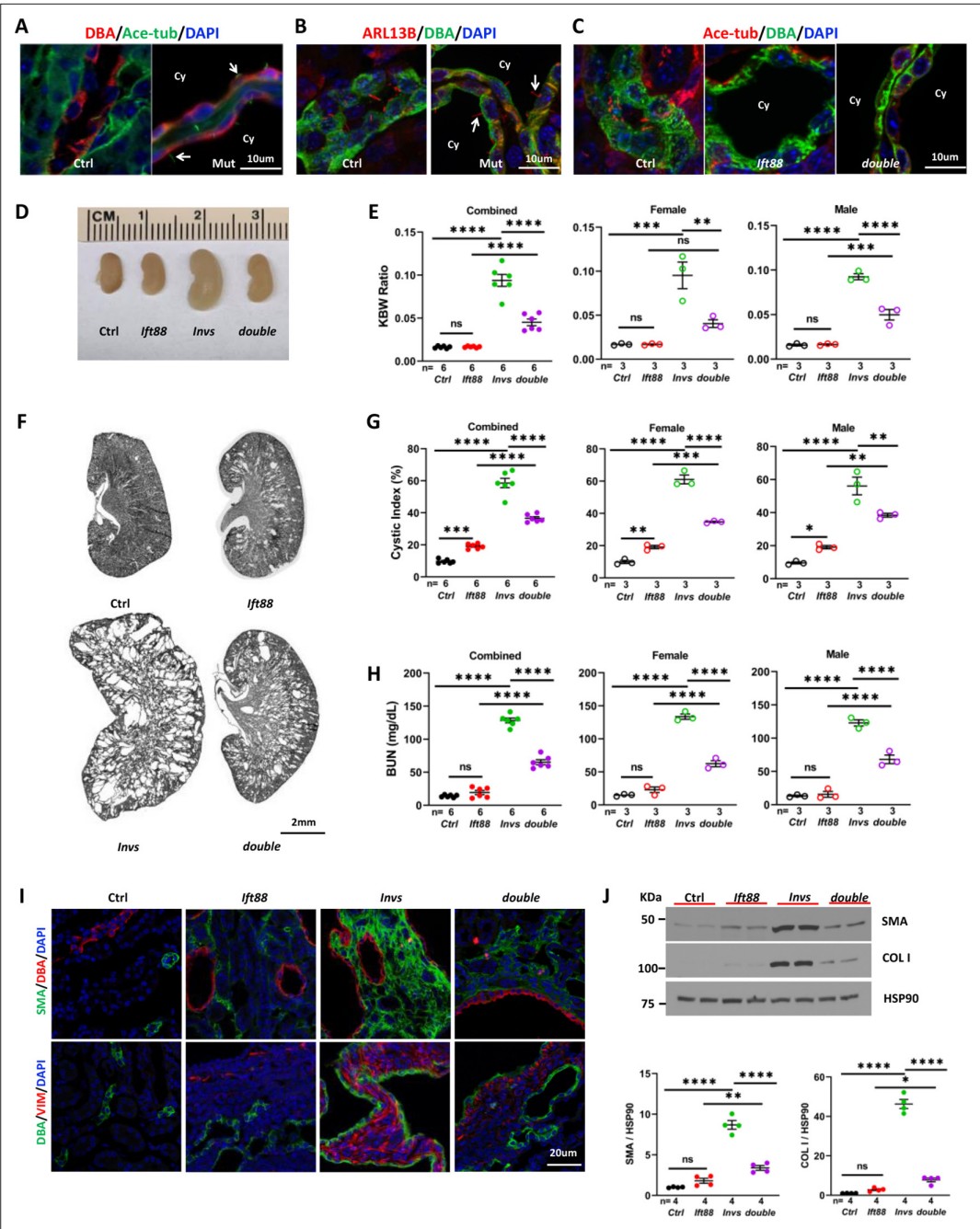

**Figure 5.** Genetic abrogation of cilia partially suppresses the phenotypes of *Invs*<sup>flox/flox</sup>;*Cdh16-Cre* mice at P21.
(**A**) *Invs* is dispensable for cilia biogenesis. Cilia (arrow) are labelled by anti-Ace-tub (green). The collecting duct is labeled by DBA (red). DAPI (blue) labels the nucleus. (**B**) *Invs* is dispensable for ciliary localization of ARL13b (red, arrow). DBA in green. DAPI in blue. (**C**) Cilia (labelled by anti-Ace-tub, red) biogenesis is disrupted in both *Ift88*<sup>flox/flox</sup>;*Cdh16-Cre* knockout and *Invs*<sup>flox/flox</sup>;*Ift88*<sup>flox/flox</sup>;*Cdh16-Cre* knockout kidney. DBA in green. (**D**) Gross kidney morphology. (**E**) KWB ratio. (**F**) HE-stained kidney sections. (**G**) Cystic index. (**H**) BUN level. (**I**) Immunostaining of the renal cortex region. DBA in red and SMA in green in upper panels. DBA in green and vimentin (VIM) in red in lower panels. DAPI in blue. (**J**) Representative western blot and quantification. COL I: Collagen I. Ctrl: age matched controls from pooled litters; *Invs: Invs*<sup>flox/flox</sup>;*Cdh16-Cre* mutant; *Ift88: Ift88*<sup>flox/flox</sup>;*Cdh16-Cre* mutant; *double: Ift88*<sup>flox/flox</sup>;*Invs*<sup>flox/flox</sup>;*Cdh16-Cre* double mutant; Ace-tub: acetylated tubulin; Cy: cyst. One-way ANOVA analysis followed by Šidák's test was performed between selected pairs. Data are presented as mean ± SEM. ns: not significant ($p > 0.05$); *: $p < 0.05$; **: $p < 0.01$; ***: $p < 0.001$; ****: $p < 0.0001$.

The online version of this article includes the following source data for figure 5:

**Source data 1.** Related to *Figure 5E, G, H and J*.

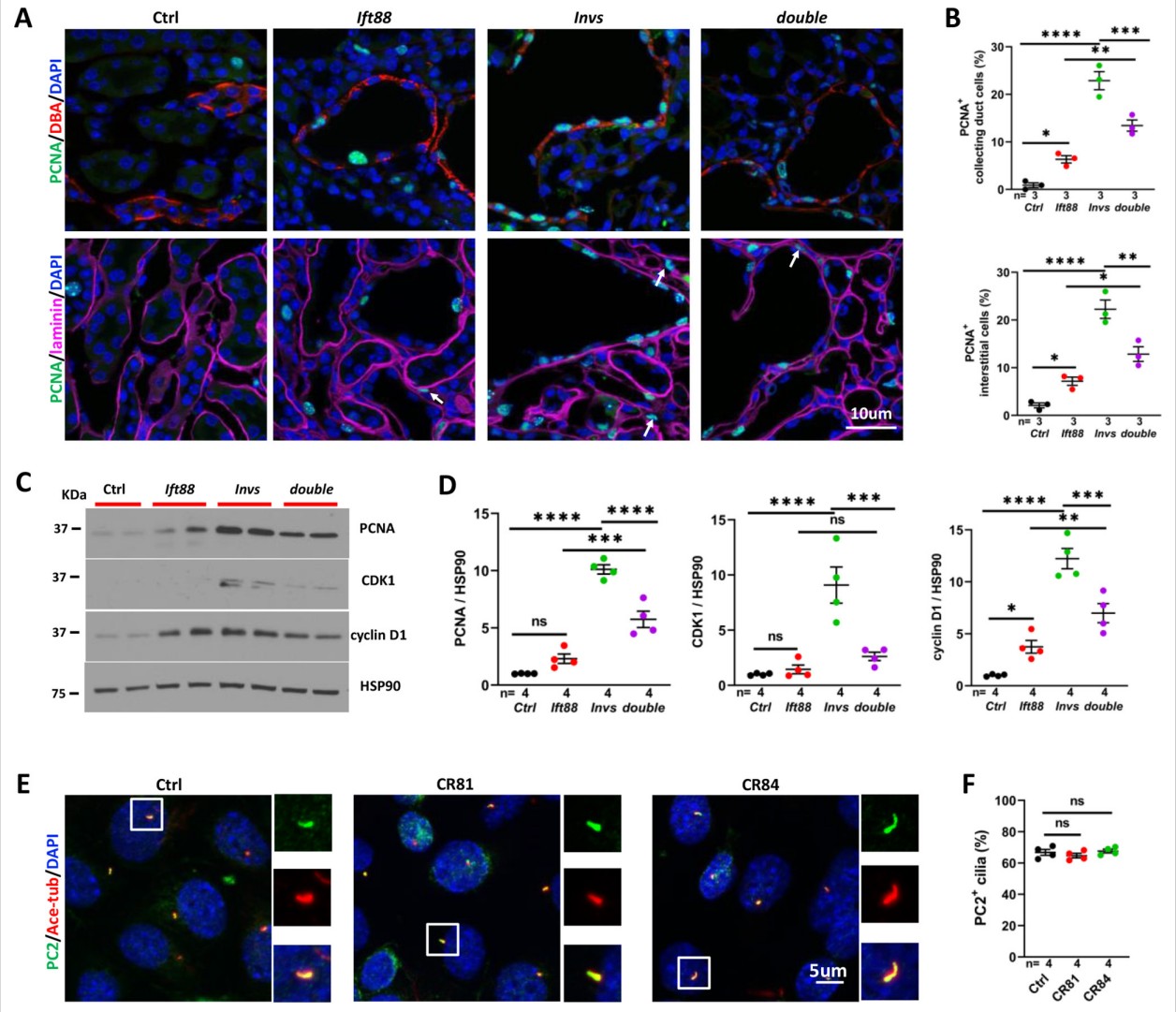

**Figure 6.** Genetic abrogation of cilia partially reduced cell proliferation in *Invs*^flox/flox^;*Cdh16-Cre* kidney at P21. (**A**) PCNA (in green) staining of the cortex region. The collecting duct is labeled by DBA in red in upper panels. Epithelial regions are encircled by laminin signal in purple in lower panels. DAPI in blue labels the nucleus. Arrows point to representative PCNA⁺ interstitial cells. (**B**) Quantification of PCNA⁺ collecting duct cells (DBA) and interstitial cells outside of regions demarcated by laminin signal. (**C**) Representative Western Blot of PCNA, CDK1, cyclin D1 and the loading control HSP90 using kidney lysates. (**D**) Quantification of signals on Western Blots. Unit 1 is defined by the level in control kidneys. (**E**) Immunostaining of PC2 (in green) in control mIMCD3 cells (Ctrl) and mutant cells (Mut) from *Invs*^-/-^ clones CR81 and CR84. Cilia are labeled by anti-acetylated tubulin (Ace-tub) in red. DAPI in blue labels the nucleus. (**F**) Quantification of PC2 positive cilia. *Invs*: *Invs*^flox/flox^;*Cdh16-Cre* mutant; *Ift88*: *Ift88*^flox/flox^;*Cdh16-Cre* mutant; *double*: *Ift88*^flox/flox^;*Invs*^flox/flox^;*Cdh16-Cre* double mutant. One-way ANOVA analysis followed by Šidák's test was performed between selected pairs. Data are presented as mean ± SEM. ns: not significant (p>0.05); *: p<0.05; **: p<0.01; ***: p<0.001; ****: p<0.0001.

The online version of this article includes the following source data and figure supplement(s) for figure 6:

**Source data 1.** Related to *Figure 6B–D and F*.

**Figure supplement 1.** The generation of *Invs*^-/-^ cells.

and B). In the *Invs*^flox/flox^;*Ift88*^flox/flox^;*Cdh16-Cre* double mutant kidney, 13.4% of DBA positive cells were PCNA positive, a significant decrease from the *Invs*^flox/flox^;*Cdh16-Cre* single mutant (**Figure 6A and B**). Further, *Ift88* deletion reduced proliferating cells in the interstitium of the *Invs*^flox/flox^;*Cdh16-Cre* kidney. In the cortex region of the control kidney, 2.1% of interstitial cells were PCNA positive. In the *Ift88*^flox/flox^;*Cdh16-Cre* mutant kidney, this number increased to 7.2% (**Figure 6A and B**). 22.3% of interstitial cells were PCNA positive in the cortex of the *Invs*^flox/flox^;*Cdh16-Cre* kidney, which reduced to 12.8% in the *Invs*^flox/flox^;*Ift88*^flox/flox^;*Cdh16-Cre* double mutant kidney (**Figure 6A and B**). To validate

the impact of *Ift88* deletion on cell proliferation, we analyzed the level of multiple proteins involved in cell proliferation by performing Western blot using P21 kidney lysates. Results showed that the level of PCNA, CDK1, a kinase required for cell cycle progression, and cyclin D1, which is essential for cell proliferation and prevents cells enter quiescence, was increased in the *Invs*^*flox/flox*^*;Cdh16-Cre* mutant kidney (**Figure 6C and D**). In the *Ift88*^*flox/flox*^*;Cdh16-Cre* mutant kidney, the level of PCNA was increased significantly by t-test but did not reach statistical significance by one-way ANOVA analysis followed by Šidák's test (**Figure 6C and D**). The level of cyclin D1, but not CDK1, was also increased (**Figure 6C and D**). Like the pattern seen by immunostaining of PCNA, the level of PCNA, CDK1 and cyclin D1 in the *Invs*^*flox/flox*^*;Ift88*^*flox/flox*^*;Cdh16-Cre* double mutant kidney was significantly decreased in comparison to the *Invs*^*flox/flox*^*;Cdh16-Cre* single mutant kidney (**Figure 6C and D**).

To investigate whether INVS functions in ciliary localization of PC2, we used mouse inner medullary collecting duct (mIMCD3) cells and generated two clones, CR81 and CR84, of *Invs*^*-/-*^ cells via CRISPR (**Figure 6—figure supplement 1**). Immunostaining with anti-PC2 and the cilia marker anti-acetylated tubulin showed no difference in ciliary localization of PC2 between mutant and control cells (**Figure 6E and F**).

Taken together, these results reveal that the phenotypes of *Invs*^*flox/flox*^*;Cdh16-Cre* mutants are partially suppressed when cilia biogenesis is disrupted genetically, similar to *Pkd2* mutants through a mechanism independent of the ciliary localization of PC2.

## The HDAC inhibitor valproic acid suppresses disease progression in *Invs*^*flox/flox*^*;Pkhd1-Cre* mice

NPHP is the most frequent monogenic cause of end stage renal disease (ESRD) in children and young adults (**Hildebrandt et al., 2009**). Currently directed therapy is lacking for NPHP. In previous studies, we found that HDACIs could partially suppress kidney cyst formation and renal fibrosis caused by inactivation of polycystins or the cilia biogenesis gene *Arl13b* in model organisms (**Cao et al., 2009**; **Li et al., 2016**). To investigate whether HDACIs could be used to suppress the phenotypes of NPHP models, we treated *Invs* mutant mice with the pan HDACI valproic acid (VPA). Since the *Invs*^*flox/flox*^*;Cdh16-Cre* model develops disease phenotypes very rapidly at the neonatal stage, posing technical challenges for drug administration, we generated *Invs*^*flox/flox*^*;Pkhd1-Cre* mice. *Pkhd1-Cre* directs Cre expression in the collecting duct starting at E14.5, later than the onset of *Cdh16-Cre* expression (**Ma et al., 2013**). In *Invs*^*flox/flox*^*;Pkhd1-Cre* mice, mutant kidneys were enlarged and cystic, with a significantly increased KBW ratio, cystic index and BUN level in both male and female mice and decreased body weight at P28 (**Figure 7A–E**). We analyzed body weight change of *Invs*^*flox/flox*^*;Pkhd1-Cre* mice in more detail and compared it to *Invs*^*flox/flox*^*;Cdh16-Cre* mice. At P28, the reduction of body weight in *Invs*^*flox/flox-*^*;Pkhd1-Cre* mice in comparison to control mice was more moderate than that in *Invs*^*flox/flox*^*;Cdh16-Cre* mice (**Figure 7—figure supplement 1**). Mutant and control mice were intraperitoneal (IP) injected with VPA from P10 to P28. Results showed that this regiment of VPA treatment decreased the size of the mutant kidney, but had no impact on body weight, in comparison to vehicle-treated mutants (**Figure 7A**, B). In addition, the KBW ratio, cystic index of kidneys and the BUN level in mutant mice were reduced significantly by VPA treatment in comparison to vehicle-treated mutants (**Figure 7B–E**). Moreover, VPA treatment led to a more moderate level of myofibroblast activation in the cortex region of the mutant kidney, as indicated by the level of SMA (**Figure 7F**). By contrast, the kidney size, KBW ratio, cystic index and BUN level were not significantly affected by the same VPA treatment in control mice, suggesting that mutant mice are selectively sensitive to VPA treatment (**Figure 7A, B, D and E**). However, the reduced body weight phenotype in mutant mice was not suppressed by VPA treatment (**Figure 7B**). We cannot rule out the possibility that the side effects of VPA contributed to weight loss in treated mice. In addition, VPA may reduce body weight through inhibiting HDAC during the growth period: the HDACI Trichostatin A was shown to inhibit adipogenesis (**Lv et al., 2021**). We then investigated the impact of VPA treatment on cell proliferation. The number of PCNA positive cells in the collecting duct labeled by DBA and the interstitium outside of tubular regions outlined by anti-laminin signal was reduced significantly by VPA treatment (**Figure 7G and H**). This result was further validated by western blot using kidney lysates. The abnormally increased level of PCNA, CDK1, and cyclin D1 in mutant kidneys was decreased significantly by VPA treatment (**Figure 7I and J**).

Combined, these results show that the HDACI VPA ameliorates the progression of multiple phenotypes in *Invs* mutants.

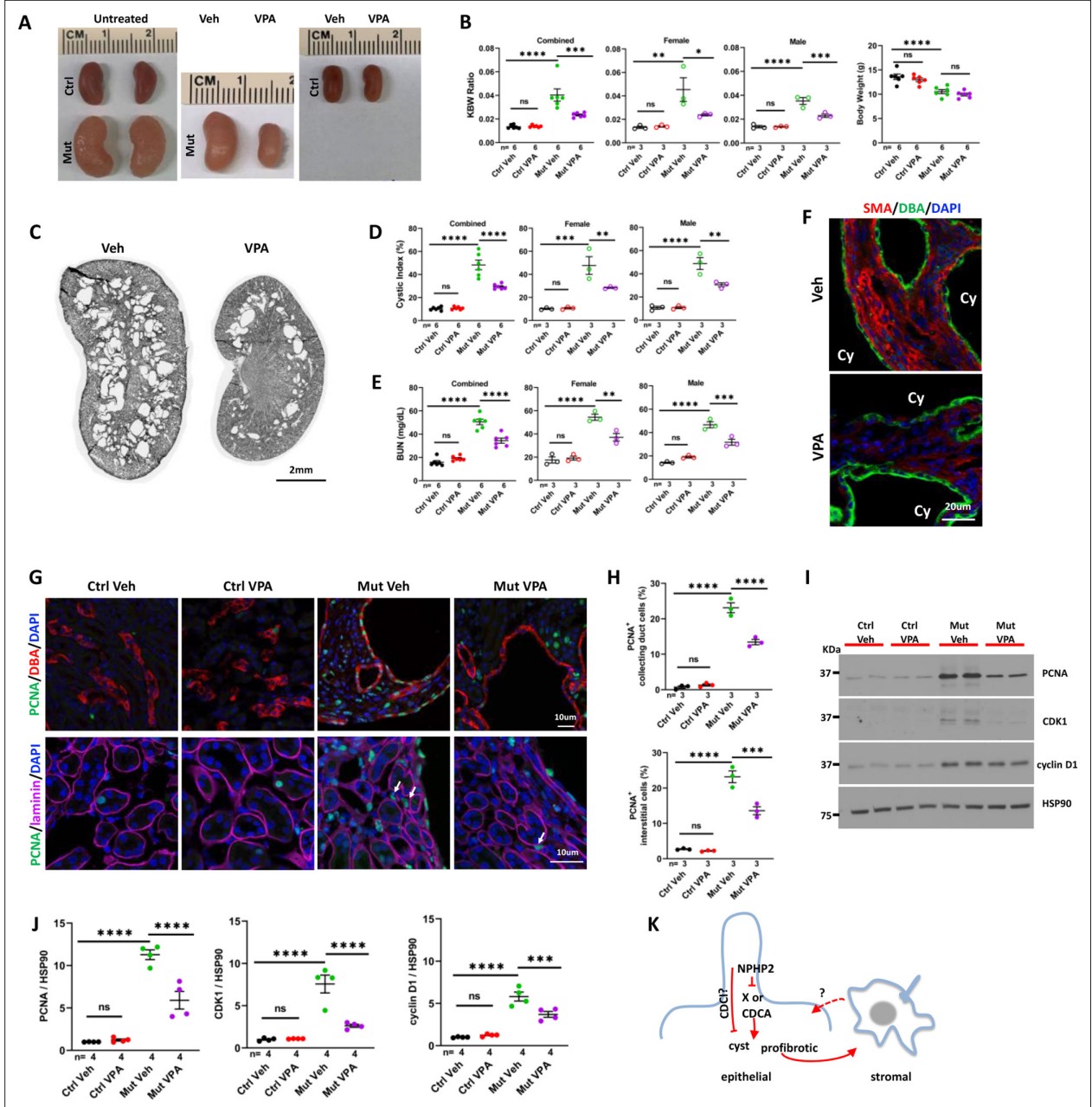

**Figure 7.** VPA partially suppresses the phenotypes of *Invs*<sup>flox/flox</sup>;*Pkhd1-Cre* mice at P28. (**A**) Gross morphology of the kidney. (**B**) KBW ratio and body weight. (**C**) HE-stained kidney section. (**D**) Cystic index. (**E**) BUN level. (**F**) SMA (red) in the cortex region of the kidney. DBA in green marks the collecting duct. DAPI in blue labels the nucleus. (**G**) Proliferating cells labelled by anti-PCNA staining (in green) in the cortex region. In upper panels, DBA in red. In lower panels, anti-laminin staining in purple. Arrows point to representative PCNA⁺ interstitial cells. DAPI in blue. (**H**) Percentage of PCNA⁺ collecting duct cells (upper panel) and interstitial cells (lower panel) in the cortex region. (**I**) Representative western blot of PCNA, CDK1, cyclin D1 and the loading control HSP90. (**J**) Quantification of signals on western blots. (**K**) A model of INVS function. INVS in epithelial cilia inhibits a pro-cystic and profibrotic pathway, which activates interstitial cells non-cell autonomously. CDCI: cilia-dependent cyst inhibiting. Cy: cyst; Ctrl: age matched control animals from pooled litters; Mut: *Invs*<sup>flox/flox</sup>;*Pkhd1-Cre* mutant; Veh: vehicle treated. One-way ANOVA analysis followed by Šidák's test was performed between selected pairs. Data are presented as mean ± SEM. ns: not significant (p>0.05); *: p<0.05; **: p<0.01; ***: p<0.001; ****: p<0.0001.

The online version of this article includes the following source data and figure supplement(s) for figure 7:

**Source data 1.** Related to *Figure 7B, D, E and H–J*.

**Figure supplement 1.** Body weight change of *Invs* models.

**Figure supplement 1—source data 1.** Related to *Figure 7—figure supplement 1A and B*.

## Discussion

Similar to infantile NPHP, mutants of the *ove* allele of *Invs*, generated by transgenic insertion of plasmid DNA in mouse, show severe cystic kidney disease and fibrosis at the neonatal stage (*Yokoyama et al., 1993*; *Mochizuki et al., 1998*; *Morgan et al., 1998*). It is possible that defective *Invs* in epithelial and stromal cells caused epithelial cysts and interstitial fibrosis, respectively, in this model. Alternatively, abnormal epithelial-stromal cross-talks, resulting from defective tissue-specific *Invs* function, might have triggered the phenotypes observed in those previous studies.

In this study, we generated a floxed allele of *Invs* and used *Cdh16-Cre* to specifically inactivate *Invs* in epithelial cells in the distal nephron and *Foxd1-Cre* to drive *Invs* deletion in the renal stroma. Significantly, inactivation of *Invs* in the distal nephron leads to cyst formation throughout the cortex and medullar region by P21, accompanied by interstitial fibrosis, recapitulating the main phenotypes of infantile NPHP. By contrast, inactivation of *Invs* in the renal stroma caused no observable phenotypes in the kidney at least up to week 8, even though the onset of *Foxd1-Cre* expression in the developing metanephros occurs at the same time as that of *Cdh16-Cre* (*Shao et al., 2002a*; *Shao et al., 2002b*; *Hum et al., 2014*; *Humphreys et al., 2010*). Combined, our results suggest that defective epithelial cells are the main driver of NPHP phenotypes in *Invs* mutants, highlighting the significance of epithelial-stromal cross-talks in the development and progression of interstitial fibrosis in this model. However, our result does not rule out functional significance of interstitial cells once a pro-cystic and fibrotic response is triggered in mutant epithelial cells.

PKD is frequently accompanied by renal fibrosis. In our previous study, we found that inactivation of the cilia biogenesis gene *Arl13b* in mouse leads to kidney cysts and fibrosis (*Li et al., 2016*). However, since the onset of fibrosis and cyst formation overlap in the $Arl13b^{flox/flox}$;*Cdh16-Cre* model, it is difficult to address whether interstitial fibrosis is solely secondary to epithelial cyst formation through, for example, mechano-stress exerted by expanding cysts. We therefore performed a detailed time course analysis of cyst formation and the fibrotic response in the $Invs^{flox/flox}$;*Cdh16-Cre* kidney. Our results showed that SMA and vimentin is already upregulated at P7, before detectable cyst formation, suggesting that the fibrotic response is triggered before cyst formation by signals from defective epithelial cells in this model. Mechano-stress on stromal cells impinged by cysts, and additional signals from cystic epithelial cells could amplify the progression of fibrosis at later stages. Interestingly, we detected an abnormal increase of proliferating cells at the pre-cystic stage in both the collecting duct and interstitium of the $Invs^{flox/flox}$;*Cdh16-Cre* kidney. An attractive model is that INVS functions to signal terminal differentiation of renal epithelial cells and *Invs* mutant cells fail to exit cell cycle to enter the quiescent stage. Currently, the precise nature of the signals from mutant cilia and epithelial cells remains unknown. Previous studies on the fibrotic response triggered by kidney injury detected upregulation of multiple developmental pathways, including TGFβ, WNT, notch, and SHH (reviewed in *Humphreys, 2018*), providing ample candidate pathways for future analysis.

Previous work suggests that INVS is specifically localized to a proximal segment of the cilium dubbed the Inversin compartment (*Shiba et al., 2009*). Super-resolution imaging revealed that INVS is localized to a novel fibril-like structure in the Inversin compartment (*Bennett et al., 2020*). Moreover, INVS interacts with multiple infantile NPHP proteins, including NPHP3, ANKS6 and NEK8; and is required for concentrating these proteins to the Inversin compartment (*Shiba et al., 2009*; *Czarnecki et al., 2015*). On the other hand, INVS is dispensable for cilia biogenesis and is also found in the plasma membrane (*Phillips et al., 2004*; *Nürnberger et al., 2002*), mitotic spindle (*Nürnberger et al., 2004*) and stress granule (*Estrada Mallarino et al., 2020*). It therefore remains possible that INVS functions outside of the cilium. To test the relationship between cilia and INVS, we removed cilia genetically in the $Invs^{flox/flox}$;*Cdh16-Cre* model by generating $Ift88^{flox/flox}$; $Invs^{flox/flox}$;*Cdh16-Cre* double knockout mouse. Significantly, *Ift88* inactivation partially reduced cell proliferation, suppressed the severity of renal cyst and fibrosis, and improved kidney function in *Invs* mutants. This result suggests that, similar to polycystins, INVS functions to inhibit a cilia-dependent pro-cystic and pro-fibrotic pathway (*Figure 7K*). Although cilia-dependent, the downstream pathway could potentially operate outside of cilia and receive parallel signals from both ciliary activity and INVS. Interestingly, genes encoding Inversin compartment components seem to form a unique functional module among the many NPHP genes. In human, mutations in Inversin compartment genes are associated with the infantile form of NPHP. In mouse, *Anks6*, *Invs*, *Nphp3,* and *Nek8* mutants develop kidney cysts, while mutants of several other NPHP genes show no obvious kidney phenotypes (*Won et al., 2011*; *Jiang*

*et al., 2008*; *Ronquillo et al., 2016*; *Morgan et al., 1998*; *Phillips et al., 2004*; *Czarnecki et al., 2015*; *Bergmann et al., 2008*; *Liu et al., 2002*; *Omran et al., 2001*). Our results exclude the ciliary localization of PC2 as a potential mechanism for the INVS mediated pathway. However, whether this pathway is related or distinct from the polycystin-mediated CDCA pathway is currently unresolved (*Figure 7K*). Understanding the Inversin compartment could provide critical clues to the molecular mechanisms linking cilia, polycystins, cyst formation, and fibrosis.

NPHP is a major cause of ESRD in children and young adults (*Hildebrandt et al., 2009*). Although the vasopressin V2 receptor antagonists OPC31260 and Tolvaptan have been shown to partially suppress phenotypic progression in *Nphp3^{pcy}* mutant mice, no directed therapy has been established for this disease (*Aihara et al., 2014*; *Gattone et al., 2003*). Epithelial cysts and macrophages have become the focus in efforts to identify novel treatment for cystic diseases. The early and severe fibrotic response in *Invs* mutants suggest that stromal cells might also serve as an important target. In this study, we show that the pan HDACI VPA reduces cell proliferation, slows down the progression of disease phenotypes and partially preserves renal function in *Invs* mutant mice. Although the expression of many genes is changed by HDACIs, clinical treatment window for these compounds exists. For example, at least four new HDACIs have been approved by FDA to treat lymphomas (*Cappellacci et al., 2020*). In addition, VPA is a drug that has been used to treat epilepsy with a known safety profile. Because of the broad target range of pan HDACIs, the precise target(s) contributing to the phenotypic improvement in *Invs* mutants remains unknown. It is attractive to speculate that a broad inhibition of multiple pathways, among them cell cycle control, by HDACIs might contribute to the effectiveness of this treatment. In the meantime, it will be informative to investigate whether the impact of HDACI treatment is HDAC type specific, thus narrowing down the list of potentially relevant targets. It is important to note that VPA could affect targets other than HDACs and testing newly approved HDACIs will provide useful insights.

## Materials and methods

**Key resources table**

| Reagent type (species) or resource | Designation | Source or reference | Identifiers | Additional information |
|---|---|---|---|---|
| Gene (*Mus musculus*) | *Invs* | NCBI | Gene ID: 16348 | MGI 1335082 |
| Strain, strain background (*M. musculus*) | C57BL/6 J | Jackson Laboratory | Cat#: 000664 | |
| Genetic reagent (*M. musculus*) | *Invs^{tm1a/+}* | EUCOMM | Embryonic stem cell clone *Invs^{tm1a}* | Yale Genome Editing Center generate *Invs^{tm1a/+}* mice |
| Genetic reagent (*M. musculus*) | *Invs^{flox/+}* | This paper | | Dr. Brueckner (Yale University) |
| Genetic reagent (*M. musculus*) | *Cdh16-Cre* | PMID:12089378, 12089379 and 23892607 | | From Stefan Somlo lab |
| Genetic reagent (*M. musculus*) | *Pkhd1-Cre* | PMID:23892607 | | From Stefan Somlo lab |
| Genetic reagent (*M. musculus*) | *Foxd1-Cre* | Jackson Laboratory | Cat#: 012463 | |
| Genetic reagent (*M. musculus*) | *Ift88^{flox/+}* | PMID:17166921, 33046531 | | From Stefan Somlo lab |
| Cell line (*M. musculus*) | mIMCD3 | ATCC | Cat#: CRL-2123 | |
| Cell line (*M. musculus*) | *Invs^{-/-}* CR81, CR84 | This paper | | Dr. Brueckner (Yale University) |
| Antibody | Anti-THP (Sheep polyclonal) | R&D Systems | Cat#: AF5175 | IF (1:500) |
| Antibody | Anti-AQP2 (Rabbit polyclonal) | Cell Signaling Technology | Cat#: 3487 S | IF (1:300) |

*Continued on next page*

*Continued*

| Reagent type (species) or resource | Designation | Source or reference | Identifiers | Additional information |
|---|---|---|---|---|
| Antibody | Anti-ARL13B (Rabbit polyclonal) | Proteintech | Cat#: 17711–1-AP | IF (1:100) |
| Antibody | Anti-acetylated tubulin (Mouse monoclonal) | Sigma-Aldrich | Cat#: MABT868; clone 6–11B-1 | IF (1:5000) |
| Antibody | Anti-α–SMA (Mouse monoclonal) | Abcam | Cat#: ab7817 | IF (1:100) |
| Antibody | Anti- vimentin (Rabbit polyclonal) | Proteintech | Cat#:10366–1-AP | IF (1:100) |
| Antibody | Anti-GFP (Mouse monoclonal) | Roche | Cat#: 11814460001 | IF (1:300) |
| Antibody | Anti-laminin (Rabbit polyclonal) | Invitrogen | Cat#: PA1-16730 | IF (1:1000) |
| Antibody | Anti-PCNA (Mouse monoclonal) | BD Biosciences Pharmingen | Cat#: 51–900205 | IF (1:100) |
| Antibody | Anti-pHH3 (mouse monoclonal) | Upstate | Cat#: 05806 | IF (1:500) |
| Antibody | Anti-INVS (Rabbit polyclonal) | Proteintech | Cat#:10585–1-AP | IF (1:100) |
| Antibody | Anti-ANKS6 (Rabbit polyclonal) | Sigma-Prestige Antibodies | Cat#: HPA008355 | IF (1:100) |
| Antibody | Anti-PC2 (Rabbit polyclonal) | PMID:10497221 | YCC2 | From Stefan Somlo Lab; IF (1:2000) |
| Antibody | Anti-CDK1 (Rabbit monoclonal) | Cell Signaling Technology | Cat#: 28439 | WB (1:1000) |
| Antibody | Anti-PCNA (Rabbit monoclonal) | Cell Signaling Technology | Cat#:13110 | WB (1:1000) |
| Antibody | Anti-cyclin D1 (Rabbit monoclonal) | Abcam | Cat#: ab134175 | WB (1:2000) |
| Antibody | Anti-HSP90 (Rabbit monoclonal) | Cell Signaling Technology | Cat#: 4877 | WB (1:3000) |
| Antibody | Anti-α-smooth muscle actin (Rabbit polyclonal) | Abcam | Cat#: ab5694 | WB (1:2000) |
| Antibody | Anti-collagen I (Rabbit polyclonal) | Proteintech | Cat#: 14695–1-AP | WB (1:2000) |
| Recombinant DNA reagent | *Invs* CRISPR/Cas9 KO (plasmid)(m) | Santa Cruz | Cat#: SC-421144 | |
| Commercial assay or kit | ProLong Gold antifade reagent with DAPI | Molecular Probes | Cat#: P36935 | |
| Commercial assay or kit | MycoAlert Mycoplasma Detction Kit | Lonza | Cat#: LT-07–118 | |
| Commercial assay or kit | Trizole reagent | Ambion | Cat#:15596018 | |
| Commercial assay or kit | SuperScript IV First-Strand Synthesis System | Invitrogen | Cat#:18091050 | |
| Commercial assay or kit | KAPA SYBR FAST Universal | KAPA Biosystems | Cat#: 07959389001 | |
| Commercial assay or kit | Protein Assay Dye Reagent Concentrate | Bio-Rad | Cat#: 500–0006 | |
| Commercial assay or kit | SuperSignal West Pico PLUS Chemiluminescent Substrate | Thermo Scientific | Cat#: 34580 | |
| Chemical compound, drug | Complete EDTA-free Protease inhibitor | Roche | Cat#: 11836170001 | |
| Chemical compound, drug | PhosSTOP phosphatase inhibitor | Roche | Cat#: 04906837001 | |
| Chemical compound, drug | 2-Propylpentanoic acid, sodium salt (VPA) | Acros Organics | Cat#: 1069-66-5 | |
| Software, algorithm | GraphPad Prism | GraphPad Prism | | |
| Software, algorithm | ImageJ | ImageJ | | |
| Other | Rhodamine DBA | Vector Laboratories | Cat#: RL-1032 | IF (1:100) |
| Other | Fluorescein DBA | Vector Laboratories | Cat#: FL-1031 | IF (1:100) |
| Other | Fluorescein LTL | Vector Laboratories | Cat#: FL1321 | IF (1:300) |

## Mouse breeding

Embryonic stem cell clones containing the *Invs^tm1a* allele were purchased from EUCOMM and injected into C57BL/6 J blastocysts to generate chimeric mice by Yale Genome Editing Center. Chimeric animals were mated to C57BL/6 J mice to generate *Invs^tm1a/+* mice. The *Invs^flox* allele was generated by crossing *Invs^tm1a/+* carrier mice with a deleter line expressing FLPe recombinase (*Rodríguez et al., 2000*) and backcrossed with C57BL/6 J for five generations. *Cdh16-Cre* and *Pkhd1-Cre* mice (*Ma et al., 2013*; *Shao et al., 2002b*) in C57BL/6 J background were kindly provided by the Somlo lab and were used to cross with *Invs^flox/+* mice to generate *Invs^flox/+*;*Cdh16-Cre* or *Invs^flox/+*;*Pkhd1-Cre* mice. *Invs^flox/flox*;*Cdh16-Cre* or *Invs^flox/flox*;*Pkhd1-Cre* mice were obtained by crossing *Invs^flox/flox* with *Invs^flox/+*;*Cdh16-Cre* mice or *Invs^flox/+*;*Pkhd1-Cre* mice. *Foxd1-Cre* purchased from Jackson Laboratory (#012463) was used to obtain *Invs^flox/+*;*Foxd1-Cre* and subsequently *Invs^flox/flox*;*Foxd1-Cre* mice. The following primers were used for genotyping:

> The floxed allele of *Invs*: *Invs flox_F* and *Invs flox_R*
> The floxed allele of *Ift88*: *Ift88 flox_F* and *Ift88 flox_R*
> Cdh16-Cre
> For carriers: *Cdh16-Cre* carrier_F and *Cdh16-Cre* carrier_R
> Non-carriers: *Cdh16-Cre* non-carrier F and *Cdh16-Cre* non-carrier_R
> *Pkhd1-Cre*: *Pkhd1-Cre*_F and *Pkhd1-Cre*_R
> *Foxd1-Cre*: following the protocol from Jackson Laboratory. Specifically, three oligos *Foxd1-Cre* wildtype Forward, *Foxd1-Cre* mutant Forward and *Foxd1-Cre* Common were used in the same reaction to amplify wild type and mutant alleles.

> Sequences of oligos are provided in *Supplementary file 1*.

## Immunostaining

Mouse kidneys were fixed by 4% PFA at 4 °C overnight, embedded in OCT and cryo-sectioned at 5 μm. Sections were incubated in blocking buffer (10% FBS in PBS with 0.1% Tween-20 (PBST)) for 1 hr at room temperature and then in primary antibody diluted with blocking overnight at 4 °C. Slides were then washed three time with PBST for 10 min each, incubated in secondary antibody diluted with blocking buffer for 1–2 hr at room temperature protected from light, followed by washing three times with PBST, mounted in ProLong Gold antifade reagent with DAPI (Molecular Probes by Life Technologies REF P36935) and imaged. The primary antibodies and lectins used are specified in Key Resources Table. Anti-PC2 (YCC2) has been described before (*Cai et al., 1999*).

## Primary culture of mouse kidney interstitial cells

A previously described protocol were followed (*Nakai et al., 2021*). Briefly, *Invs^flox/flox*;*Foxd1Cre* and sibling control mice were sacrificed at P12. Kidneys were harvested and the fibrous capsule was removed using forceps. Kidney tissue was minced with razor blade in 1 mL 0.25% trypsin (Gibco 25200–056) on a Petri dish at room temperature and incubated at 37 °C for 30–45 min. Trypsin activity was quenched by adding 4 mL DMEM (Gibco 11965–092) with Antibiotic-Antimycotic (Gibco 15240–062) and 10% FBS (Gibco 16140–071) per dish. Tissues were broken up by pipetting 10–20 times. Cell suspension was filtered through a 100 μm sterile cell strainer (Fisher Scientific #22363549) and 40 μm sterile cell strainer (Fisher Scientific #22363549). Cell suspension was centrifuged at 1000 *g* for 5 min and supernatant was removed. Cell pellet was resuspended in media. Cells were seeded on Microscope Cover Glasses (Glaswarenfabrik Kari Hecht GmbH& Co. 92100100030) placed in cell culture dishes (FALCON 353001). Cells were allowed to grow to confluency and subjected to serum starvation for 24 hr before fixation. Cells were fixed with MeOH at –20 °C for immunostaining of INVS and with 4% PFA for immunostaining of ANKS6.

## Cystic index measurement

Cystic index was measured as previously published (*Shibazaki et al., 2008*). Briefly, two sections were analyzed for each animal. Hematoxylin/eosin-stained sections were scanned using the scan slide module in Metamorph v.7.1 acquisition software (Universal Imaging). Cystic and non-cystic areas were measured by the Region Statistics feature in Metamorph. Cystic index is defined as cystic area to total kidney area in each section.

## RT-qPCR

Mouse kidneys were lyzed in the Trizole reagent (Ambion by Life Technologies REF 15596018) and total RNA was extracted following the manufacturer's instructions. Total RNA was used to synthesize cDNA using the SuperScript IV First-Strand Synthesis System (Invitrogen by Thermo Fisher Scientific REF 18091050). QPCR was performed using the KAPA SYBR FAST Universal (REF 07959389001) and BIO-RAD CFX96 Real-time System. Oligos used are listed in *Supplementary file 1*. *Gapdh* was used as internal control (*Jonassen et al., 2008*).

## Generation of *Invs*⁻/⁻ cells through CRISPR-Cas9

mIMCD3-cells were purchased from ATCC (CRL-2123) and used before passage 25. MycoAlert Mycoplamsma Detction Kit (Lonza, REF LT-07–118) was used to verify that these cells were negative for mycoplasma. RNA-seq validated that the cells were of murine origin. The morphology and karyotype of the cells were also consistent with known profiles of mIMCD3 cells (epithelial, near triploid). A pool of three plasmids for CRISPR/Cas9 knockout of *Invs* (SC-421144) was purchased from Santa Cruz Biotechnology and used following the manufacturer's protocol. These plasmids direct the expression of the reporter GFP and sgRNAs against the 5' side of *Invs*. mIMCD3 cells were electroporated and electroporation efficiency was monitored by GFP signal. Single colonies were selected by puromycin treatment. The target regions were PCR amplified and subjected to fragment analysis. Clones with changed size of target regions were further analyzed by RT-PCR using 5'-GAACACCACTTATGTACTGT GTG-3' and 5'-GATGCTGCAAAAACACTTTGAC-3'. PCR products were cloned into the TA TOPO vector and sequenced. Clones CR81 and CR84 contains indels that lead to frameshift and premature stop codon after the first 105 and 90 amino acid of INVS, respectively.

## Quantification of PCNA and pHH3-positive cells

At least three randomly selected field in the cortex region and at least 1000 DBA positive or interstitial cells outside of regions encircled by Laminin were evaluated per animal. Three animals were evaluated for each condition.

## Protein extraction, SDS-PAGE, western blot, and quantification

Mouse kidney was homogenized in ice-cold Tris lysis buffer (50 mM Tris-HCl, pH 8.0, 150 mM NaCl, 1% NP40 and freshly added 0.1 mM DTT) with Complete EDTA-free Protease inhibitor (Roche, 11836170001) and PhosSTOP phosphatase inhibitor (Roche, 04906837001). Lysates were cleared by centrifugation at 12,000 *g* at 4 °C for 5 min. Protein concentration was measured using Protein Assay Dye Reagent Concentrate (Bio-Rad #500–0006). Equal amounts of total protein were loaded, separated on Mini-Protean TGX Precast Gels (Bio-Rad) and transferred to PVDF membranes. Membranes were incubated with 5% Non-fat milk in TBST (TBS +0.1% Tween-20) at room temperature for 1 hr and then incubated with primary antibodies at 4 °C overnight, followed by HRP-conjugated secondary antibodies (Jackson ImmunoResearch, 1:5000 in 5% Non-fat milk). Signal was detected using Super-Signal West Pico PLUS Chemiluminescent Substrate (34580, Thermo Scientific). Western blot signals from four groups of mouse kidneys were quantified using ImageJ IntDen software. Densitometric ratios are relative to HSP90 on the same blot. Fold change is shown relative to the mean of the ratio in the control samples on each blot, which is set to a value of 1.0. The antibodies used are specified in Key Resources Table.

## VPA reatment

VPA was purchased from Acros Organics (CAS: 1069-66-5). Daily IP injection was performed from P10 to P28. The VPA treatment group received 200 mg/kg VPA in 15% DMSO and 85% saline, whereas the vehicle control group received the same volume of 15% DMSO and 85% saline without VPA.

## Statistical analyses

Unpaired t-test between mutants and control animals (*Figures 1–4*, *Figure 4—figure supplement 1* and *Figure 7—figure supplement 1*) and one-way ANOVA analysis followed by Šidák's test between selected pairs among multiple groups (*Figures 5–7*) were performed using GraphPad Prism 9.2.0. $p=0.05$ was used as the threshold for statistical significance. All data are presented as mean ± SEM.

## Acknowledgements

We thank Chia-Ling Hsieh for karyotyping mIMCD3 cells, members of the Brueckner laboratory, Somlo laboratory and Sun laboratory for helpful discussions; S Somlo for insightful feedbacks; A Cox for critical reading of the manuscript; the Somlo lab for *Ift88*<sup>flox/+</sup>, *Cdh16-Cre* and *Pkhd1-Cre* mice and anti-PC2, S Mentone for histology assistance; and D Lonnette in George M O'Brien Kidney Center at Yale for BUN analysis. Funding: This work was supported by National Institute of Health grants R01DK113135, R01HD093608 (to Dr Sun) and R35HL145249 (to Dr Brueckner). The George M O'Brien Kidney Center at Yale was supported by P30 DK079310 from NIH.

## Additional information

### Funding

| Funder | Grant reference number | Author |
| --- | --- | --- |
| National Institutes of Health | R01DK113135 | Zhaoxia Sun |
| National Institutes of Health | R01HD093608 | Zhaoxia Sun |
| National Institutes of Health | R35HL145249 | Martina Brueckner |

The funders had no role in study design, data collection and interpretation, or the decision to submit the work for publication.

### Author contributions

Yuanyuan Li, Data curation, Formal analysis, Investigation, Methodology, Writing - original draft, Project administration, Writing - review and editing; Wenyan Xu, Svetlana Makova, Data curation, Formal analysis, Investigation; Martina Brueckner, Conceptualization, Supervision, Funding acquisition, Project administration; Zhaoxia Sun, Conceptualization, Formal analysis, Supervision, Funding acquisition, Writing - original draft, Project administration

### Author ORCIDs

Zhaoxia Sun http://orcid.org/0000-0002-2307-7719

### Ethics

All mouse experiments were performed in Yale University School of Medicine in accordance with Yale University Institutional Animal Care and Use Committee guidelines. Protocols were approved by Yale University Institutional Animal care and Use Committee (Protocol number: 2022-11546).

### Decision letter and Author response

Decision letter https://doi.org/10.7554/eLife.82395.sa1
Author response https://doi.org/10.7554/eLife.82395.sa2

## Additional files

### Supplementary files
- MDAR checklist
- Supplementary file 1. Sequences of oligos used in this study.

### Data availability

No large scale datasets generated. Data analyzed can be found in source data files for Figures 1–7, Figure 4—figure supplement 1 and Figure 7—figure supplement 1.

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
