## [Editor Report]

Germline inactivation of NPHP2, which encodes a protein that localizes to the transition zone at the base of the primary cilium, results in infantile kidney cysts and fibrosis. In this study, the authors provide solid evidence that increased cell proliferation and fibrosis precede cyst formation in Nphp-2 mouse models, that mutant renal epithelial cells are responsible for the phenotype, and that genetic inhibition of ciliogenesis in this model reduces disease severity. They also show that valproic acid, a drug that affects a number of cellular targets and is used to treat other human conditions, slows disease progression.

---

## [Decision Letter]

**Decision letter after peer review:**

Thank you for submitting your article "Inactivation of *Nphp2* in renal epithelial cells drives infantile nephronophthisis like phenotypes in mouse" for consideration by *eLife*. Your article has been reviewed by 3 peer reviewers, one of whom is a member of our Board of Reviewing Editors, and the evaluation has been overseen by a Reviewing Editor and Martin Pollak as the Senior Editor. The reviewers have opted to remain anonymous.

Essential revisions:

1) Please provide an explanation for why the Nphp2 mutants in Figure 7, regardless of treatment status, have body weights that are significantly lower than the controls, with treated mutants even trending lower than their untreated mutant counterparts.

2) Please provide more details on how you controlled for possible genetic background effects as this factor could be an important confounder for the interpretation of your genetic interaction and valproic acid treatment studies.

3) Please acknowledge that the mechanism by which VPA is exerting its effects is not certain in this study as VPA has been reported to have numerous pharmacologic effects and targets and your study provides no direct evidence in support of any one specifically.

4) Please explain how your double knock-out studies "support a significant role of cilia in Nphp2 function in vivo" and can exclude that ciliary activity is operating in an Nphp2-independent, parallel fashion that modulates some common downstream pathways.

5) While most of the issues can either be addressed by more explanation or editing, the reviewers have asked for more data to support several of your conclusions. This could be in the form of additional data from timepoints later than 28 days to support your conclusions that Nphp2 loss in stromal cells is not important to cystic kidney disease or kidney fibrosis, since late cyst formation has been reported in mice with stromal inactivation of Pkd1. Alternatively, you can compare the effects of global activation of Nphp2 to epithelial cell-specific inactivation of Nphp2 to support your conclusion that stromal loss of Nphp2 has no effect on early, severe cystic disease.

6) Although this is not required, it would strengthen the paper to include more timepoints and data supporting their conclusion that fibrosis precedes cyst development in the Nphp2 epithelial cell ko model.

*Reviewer #1 (Recommendations for the authors):*

General comment: This is a solid study but its results and impact are quite modest. The study's impact would be greatly strengthened if it provided some insight into the nature of the mutant epithelial cell-"normal" stomal cell signaling, how impaired ciliogenesis ameliorates disease (beyond the reduction in cellular proliferation and fibrosis), or a more mechanistic understanding of the effects of VPA.

*Reviewer #2 (Recommendations for the authors):*

I have a few comments that might be addressed or discussed further to improve the interpretations of underlying mechanisms of Nphp2 function in renal homeostasis.

First, as the authors allude to the nature of the NPHP-regulated cilia-dependent pathway is related or distinct from the polycystin-mediated CDCA pathway is currently unresolved. The lack of PC2 trafficking in Nphp2 mutants suggests that the CDCA pathway might not at all be activated in the Nphp2 mutants. Rather NPHP2 could be functioning independently in regulating cystogenesis from PKD-regulated CDCA. Testing double mutants of Nphp2 with Pkd1/2 might resolve this issue.

Second, an independent interpretation of Nphp2 lack-induced cystogenesis being reduced partially from ciliary disruption is that the lack of Nphp2 or cilia regulates independent pathways with relation to fibrosis and epithelial regulation. Late assessment of phenotypes such as epithelial and stromal proliferation by the authors shows partial restorative effects. In fact, the authors show partial inhibition of stromal proliferation in Nphp2 loss from epithelial lack of cilia which is very interesting. As an early indicator of Nphp2 inactivation is stromal fibrosis and proliferation, does parallel cilia disruption prevent such early phenotypes? Does cilia disruption itself cause fibrosis at earlier stages? Such early assessment of phenotypes might be more insightful in the cilia single mutants and double mutants.

Third, as NPHP2 interacts with multiple infantile NPHP proteins, including NPHP3, ANKS6, and NEK8 in the inversin compartment, one way of ruling out if Nphp2 function is in cilia, would be to compare/discuss phenotypes with other mutants that disrupt these interacting genes. As NPHP2 is localized to a novel fibril-like structure in the inversin compartment in cilia, the elephant in the room is the lack of our current understanding of the inversin compartment's role in the ciliary composition. Such understanding could provide important mechanistic clues and might also provide insights into the role of inversin compartment in regulating L-R asymmetry.

*Reviewer #3 (Recommendations for the authors):*

Using a newly established floxed model of Nphp2, the authors show convincing data that Nphp2 loss in the kidney epithelium is critical to kidney cyst formation and fibrosis. In addition, the authors show compelling data that cilia loss in the setting of epithelial Nphp2 loss reduces kidney cyst severity. This cilia-dependent idea of PKD has been previously published in the setting of ADPKD, but remained unexplored for NPHP. No additional, or novel mechanistic insight is shown in the manuscript. The authors also suggest that Nphp2 loss in stromal cells is not important to cystic kidney disease or kidney fibrosis, but that data is underdeveloped. Lastly, the authors perform a preclinical trial using a broad-spectrum HDAC inhibitor, in their model, which shows efficacy in slowing disease. This experiment seems disconnected from the rest of the paper.

Ksp-cre model characterization: (1) It would be important to compare the Ksp-cre results to a global knockout of Nphp2 (CAGGc-cre) in terms of PKD severity. The authors stress the possible importance of epithelial-stromal communication. To better define this a comparison of epithelial versus global knock-out is needed. (2) It would be helpful to have a Kaplan-Meyer curve of the model. (3) Could the authors comment on what percentage of DBA or THP-positive tubules are cystic in the model? (4) The data supporting that fibrosis precedes cysts is underdeveloped. The SMA staining at P7 seems more intense than at P14 which is surprising (P14 and later is not quantified – cystic versus non-cystic regions), and vimentin is not analyzed at the later stages. (5) The data that epithelial Nphp2 loss results in higher proliferation in the interstitium are interesting; however, the authors provide no experiments/speculations on why that may be. Further, these analyses are done only at P7, and it is unclear how it pertains to stages when cysts develop/are present. And, it is unclear why for all other figures examining proliferation, the authors performed PCNA staining and western blotting of essential proteins, but not for this one.

Foxd1-cre model characterization: (1) The characterization of this model is underdeveloped. Animals need to be aged much longer to assure no phenotype is observed. (2) The experiments performed in Figure 4I are not intuitive. Based on the mouse model shown in Figure 1A it is unclear where eGFP expression should come from. (3) While this is intrinsically an issue with the Foxd1-cre model, the observation that was made that only 60% of all stromal cells lost Nphp2 (Figure 4J) is concerning to make such a strong conclusion that NPHP2 in the stromal compartment is not important to the NPHP phenotype (fibrosis or cysts).

NPHP2 and Cilia: (1) While the finding that the CDCA may be critical to NPHP, as well as ADPKD, is novel, the provided data is only descriptive. It would improve the paper drastically if the authors could provide data suggesting whether NPHP2 functions in the same pathway as PC1 or PC2 or a different one. (2) For their cell analyses, the authors should show successful loss of NPHP2 in their CRISPR clones as well as also analyze PC1. (3) Fibrosis was not analyzed.

HDAC preclinical trial: This data is not connected to anything else within the manuscript. It utilized a different cre than for all the other experiments and is not mechanistically linked to the story of whether NPHP2 function in epithelial or stromal cells is important to pathogenesis. Unless the authors can link the stories better or highlight why the HDAC function is important to the observations, this reviewer would suggest removing Figure 7.

---

## [Author Response]

Essential revisions:1) Please provide an explanation for why the Nphp2 mutants in Figure 7, regardless of treatment status, have body weights that are significantly lower than the controls, with treated mutants even trending lower than their untreated mutant counterparts.

We examined the time course of body weight change more carefully and added Figure 7—figure supplement 1 to present the results. Although *Invs^flox/flox^;Pkhd1-Cre* mice displayed reduced body weight at P28 in comparison to controls, this reduction was more moderate than that of *Invs^flox/flox^;Cdh16-Cre* mice (Figure 7—figure supplement 1A). Notably, the trend of body weight difference started at around P21 in both *Invs^flox/flox^;Pkhd1-Cre* and *Invs^flox/flox^;Cdh16-Cre* mice, coinciding with weaning (Figure 7—figure supplement 1B). It is possible that mutants with compromised kidney function were less capable to thrive and gain weight at around this transition time. In terms of VPA treatment, body weight trended down in both wild type and mutant mice subjected to the treatment, although the difference did not reach statistical significance (Figure 7B). We cannot rule out the possibility that the side effects of VPA contributed to weight loss in treated mice. In addition, VPA may reduce body weight through inhibiting HDAC: the HDAC inhibitor Trichostatin A was shown to inhibit adipogenesis (PMID: 34232916) and 4-hexylresorcinol, another HDAC inhibitor, reduced body weight in treated rats (PMID: 34445640). To include the additional data and references, we added the following in the Results section:

"We analyzed body weight change of *Invs^flox/flox^;Pkhd1-Cre* mice in more detail and compared it to *Invs^flox/flox^;Cdh16-Cre* mice. At P28, the reduction of body weight in

*Invs^flox/flox^;Pkhd1-Cre* mice in comparison to control mice was more moderate than that in *Invs^flox/flox^;Cdh16-Cre* mice (Figure 7—figure supplement 1)."

“However, the reduced body weight phenotype in mutant mice was not suppressed by VPA treatment (Figure 7B). We cannot rule out the possibility that the side effects of VPA contributed to weight loss in treated mice. In addition, VPA may reduce body weight through inhibiting HDAC during the growth period: the HDACI Trichostatin A was shown to inhibit adipogenesis (51)**."**

2) Please provide more details on how you controlled for possible genetic background effects as this factor could be an important confounder for the interpretation of your genetic interaction and valproic acid treatment studies.

In Figure 5 (genetic interaction), all mice are in C57BL/6J background. In this experiment, multiple genotypes (i.g. *Invs^flox/flox^;Cdh16-Cre*, *Invs^flox/flox^*;*Ift88^flox/flox^;Cdh16-Cre* and *Ift88^flox/flox^;Cdh16-Cre*) were analyzed. Although control and mutant littermates were used, nonlittermates had to be included in some cases because of the limited number of animals per litter and low yield of desired genotypes. Littermate status is now highlighted by colors in the data tables of Figure 5 source data. See Author response table 1 for KBW ratio.

**Author response table 1. sa2table1:** 

	Group A	Group B	Group C	Group D
	* **Ctrl** *	* **Ift88** *	* **Invs** *	* **double** *
**1**	0.015800000	0.015878788	0.102857143	0.049194030
**2**	0.017560000	0.016804124	0.099063291	0.056283951
**3**	0.016694444	0.017709302	0.088949153	0.035397260
**4**	0.015176471	0.015256757	0.066209677	0.038000000
**5**	0.014500000	0.016518519	0.116844444	0.055061224
**6**	0.017973333	0.017322034	0.089321429	0.036822785

In Figure 7 (VPA), all mice are in C57BL/6J background. Because of the limited number of animals per litter and the need to subject each genotype to VPA and vehicle treatment, nonlittermates had to be included in some cases. Littermate status is now labeled by highlight colors in the data tables of Figure 7 source data. The table for KBW ratio is pasted in Author response table 2.

**Author response table 2. sa2table2:** 

	Group A	Group B	Group C	Group D
	Ctrl Veh	Ctrl VPA	Mut Veh	Mut VPA
1	0.012396552	0.014127820	0.037855556	0.020752688
2	0.015106870	0.013128000	0.065266055	0.022039604
3	0.012455882	0.013617391	0.038261307	0.024288889
4	0.012109489	0.015356589	0.033000000	0.025181818
5	0.015722628	0.013800000	0.037882353	0.025700000
6	0.012873418	0.012978723	0.029415929	0.022441176

We added more detailed description of genetic background in the Materials and methods section. Littermate status is now also indicated in figure legends.

3) Please acknowledge that the mechanism by which VPA is exerting its effects is not certain in this study as VPA has been reported to have numerous pharmacologic effects and targets and your study provides no direct evidence in support of any one specifically.

We agree that this is an important point to clarify. We added the following text to the Discussion section:

"It is important to note that VPA could affect targets other than HDACs and testing newly approved HDACIs will provide useful insight."

4) Please explain how your double knock-out studies "support a significant role of cilia in Nphp2 function in vivo" and can exclude that ciliary activity is operating in an Nphp2-independent, parallel fashion that modulates some common downstream pathways.

Our results and model do not exclude the possibility that INVS and ciliary activity feed into a common downstream pathway, i.e., a cilia-dependent cyst-activating pathway could operate outside of cilia. We added "Although cilia-dependent, the downstream pathway could potentially operate outside of cilia and receive parallel signals from both ciliary activity and Invs." to Discussion to clarify and reflect the results and model more accurately.

5) While most of the issues can either be addressed by more explanation or editing, the reviewers have asked for more data to support several of your conclusions. This could be in the form of additional data from timepoints later than 28 days to support your conclusions that Nphp2 loss in stromal cells is not important to cystic kidney disease or kidney fibrosis, since late cyst formation has been reported in mice with stromal inactivation of Pkd1. Alternatively, you can compare the effects of global activation of Nphp2 to epithelial cell-specific inactivation of Nphp2 to support your conclusion that stromal loss of Nphp2 has no effect on early, severe cystic disease.

Regarding comparing epithelial-specific and global knockout models, for an interpretable comparison, it is essential that the stage and knockout efficiency in epithelial cells are matched between the two models. However, *Cdh16-Cre* is expressed in the distal nephron specifically, sparing epithelial cells in other segments, while epithelial cells in all segments would be affected by *Cagg-Cre*. In addition, global knockout of *Invs* leads to peri-natal lethality. Inducible *Cagg-Cre* could potentially be used to bypass earlier functional requirements. But matching stage and knockout efficiency in renal epithelial cells between *Cdh16-Cre* and inducible *CaggCre* mediated knockout remains challenging. These factors make a direct comparison problematic. Finally, our results revealed the role of defective epithelial cells in triggering the phenotypes but did not rule out a role of interstitial cells once abnormal signaling is initiated in epithelial cells. To clarify this point, we added "However, our result does not rule out functional significance of interstitial cells once a pro-cystic and fibrotic response is triggered in mutant epithelial cells." to the Discussion section.

In addition, we followed the first suggested approach and expanded our analysis to 8-week-old mice. We now show that *Invs^flox/flox^;Foxd1-Cre* mice displayed normal kidney weight, KBW ratio, kidney function and histology at P56. By comparison, *Invs^flox/flox^;Cdh16-Cre* kidneys were already cystic at P14. The new P56 results are presented in Figure 4—figure supplement 1 and described in the Results section. We also added "up to the young adult stage" to make our conclusion regarding the lack of renal phenotypes in interstitial KO of *Invs* more precise.

6) Although this is not required, it would strengthen the paper to include more timepoints and data supporting their conclusion that fibrosis precedes cyst development in the Nphp2 epithelial cell ko model.

Results at P7, 9, 14 and later timepoints were presented in Figure 2. At P14, *Invs^flox/flox^;Cdh16Cre* kidneys were already cystic. At P7, the increase of SMA and vimentin was modest. We therefore analyzed kidneys at P9, a timepoint between P7 and P14, in more detail. Results showed normal KBW ratio and normal kidney morphology at P9. However, vimentin signal was increased, in addition to the increase of SMA signal detected previously. A caveat of this result is that mild tubule dilation could be detected at P9. We added the new results in Figure 2 and the Results section.

Reviewer #2 (Recommendations for the authors):I have a few comments that might be addressed or discussed further to improve the interpretations of underlying mechanisms of Nphp2 function in renal homeostasis.First, as the authors allude to the nature of the NPHP-regulated cilia-dependent pathway is related or distinct from the polycystin-mediated CDCA pathway is currently unresolved. The lack of PC2 trafficking in Nphp2 mutants suggests that the CDCA pathway might not at all be activated in the Nphp2 mutants. Rather NPHP2 could be functioning independently in regulating cystogenesis from PKD-regulated CDCA. Testing double mutants of Nphp2 with Pkd1/2 might resolve this issue.

We agree with the reviewer that testing double mutants of *Invs* and *Pkd1/2* will provide important insight. Although this is a research direction we are interested in pursuing, we feel it ultimately is outside the scope of this manuscript.

Second, an independent interpretation of Nphp2 lack-induced cystogenesis being reduced partially from ciliary disruption is that the lack of Nphp2 or cilia regulates independent pathways with relation to fibrosis and epithelial regulation. Late assessment of phenotypes such as epithelial and stromal proliferation by the authors shows partial restorative effects. In fact, the authors show partial inhibition of stromal proliferation in Nphp2 loss from epithelial lack of cilia which is very interesting. As an early indicator of Nphp2 inactivation is stromal fibrosis and proliferation, does parallel cilia disruption prevent such early phenotypes? Does cilia disruption itself cause fibrosis at earlier stages? Such early assessment of phenotypes might be more insightful in the cilia single mutants and double mutants.

We agree with the reviewer that analyzing double mutants at earlier timepoints potentially could lead to interesting findings. Technically, cilia abrogation due to the loss of IFT genes is a slow process (PMID: 23892607). Our current system, in which knockout of *Invs* and *Ift88* was driven by the same *Cdh16-Cre*, was not particularly suitable for this analysis as cilia would persist at earlier timepoints. In the revision, we analyzed markers of myofibroblast activation and fibrosis via immunostaining and Western Blot at P21 and showed a reduction of these markers in double mutants. In addition, our results showed that at P21, the increase of SMA, vimentin and collagen I was more moderate in *Ift88* mutants than in *Invs* mutants. The new results were added to Figure 5 and described in Results.

Third, as NPHP2 interacts with multiple infantile NPHP proteins, including NPHP3, ANKS6, and NEK8 in the inversin compartment, one way of ruling out if Nphp2 function is in cilia, would be to compare/discuss phenotypes with other mutants that disrupt these interacting genes. As NPHP2 is localized to a novel fibril-like structure in the inversin compartment in cilia, the elephant in the room is the lack of our current understanding of the inversin compartment's role in the ciliary composition. Such understanding could provide important mechanistic clues and might also provide insights into the role of inversin compartment in regulating L-R asymmetry.

Following reviewer suggestion, we added "Interestingly, genes encoding Inversin compartment components seem to form a unique functional module among the many NPHP genes. In human, mutations in Inversin compartment genes are associated with the infantile form of NPHP. In mouse, *Anks6*, *Invs*, *Nphp3* and *Nek8* mutants developed kidney cysts, while mutants of several other NPHP genes show no obvious kidney phenotypes" to Discussion.

Reviewer #3 (Recommendations for the authors):Using a newly established floxed model of Nphp2, the authors show convincing data that Nphp2 loss in the kidney epithelium is critical to kidney cyst formation and fibrosis. In addition, the authors show compelling data that cilia loss in the setting of epithelial Nphp2 loss reduces kidney cyst severity. This cilia-dependent idea of PKD has been previously published in the setting of ADPKD, but remained unexplored for NPHP. No additional, or novel mechanistic insight is shown in the manuscript. The authors also suggest that Nphp2 loss in stromal cells is not important to cystic kidney disease or kidney fibrosis, but that data is underdeveloped. Lastly, the authors perform a preclinical trial using a broad-spectrum HDAC inhibitor, in their model, which shows efficacy in slowing disease. This experiment seems disconnected from the rest of the paper.Ksp-cre model characterization: (1) It would be important to compare the Ksp-cre results to a global knockout of Nphp2 (CAGGc-cre) in terms of PKD severity. The authors stress the possible importance of epithelial-stromal communication. To better define this a comparison of epithelial versus global knock-out is needed.

As discussed above, it is essential for an interpretable comparison but technically challenging to match knockout timing and efficiency in epithelial cells between a global knockout model and a Cdh16-Cre model.

(2) It would be helpful to have a Kaplan-Meyer curve of the model.

We agree that a Kaplan-Meyer curve could provide useful information. However, our current animal protocol does not allow a death as endpoint experiment.

(3) Could the authors comment on what percentage of DBA or THP-positive tubules are cystic in the model?

We counted close to 200 tubules for each marker on P21 kidney sections. 91.9% DBA positive tubules were cystic and 39.3% THP positive tubules were cystic.

(4) The data supporting that fibrosis precedes cysts is underdeveloped. The SMA staining at P7 seems more intense than at P14 which is surprising (P14 and later is not quantified – cystic versus non-cystic regions), and vimentin is not analyzed at the later stages.

The increase of SMA staining is more modest at P7 than at P14 (Figure 2, please compare B and F). We have now performed immunostaining for vimentin at more timepoints (P7, 9, 14 and 21) and results were added to Figure 2.

(5) The data that epithelial Nphp2 loss results in higher proliferation in the interstitium are interesting; however, the authors provide no experiments/speculations on why that may be. Further, these analyses are done only at P7, and it is unclear how it pertains to stages when cysts develop/are present. And, it is unclear why for all other figures examining proliferation, the authors performed PCNA staining and western blotting of essential proteins, but not for this one.

P7 is a precystic stage and we used it to detect defects that are not secondary to cyst formation. Increased proliferation at an earlier stage could potentially contribute to overt cyst at a later stage and increased proliferation was detected at P21, a cystic stage (Figure 6A, B). Regarding Western Blotting, we developed these assays for the analysis of double mutants and VPA treatment, after the completion of the characterization of P7 single mutants. Moreover, we feel the difference shown by immunostaining and quantification of PCNA and phosphohistone H3 positive cells was already convincing (Figure 3).

Foxd1-cre model characterization: (1) The characterization of this model is underdeveloped. Animals need to be aged much longer to assure no phenotype is observed.

We now show that *Invs^flox/flox^;Foxd1-Cre* mice show normal kidney weight, kidney/body weight ratio, kidney function and histology at P56. These results were presented in Figure 4—figure supplement 1 and described in the Results section.

(2) The experiments performed in Figure 4I are not intuitive. Based on the mouse model shown in Figure 1A it is unclear where eGFP expression should come from.

We apologize for not making this clear. Cre is tagged with eGFP in *Foxd1-Cre*. We changed eGFP to "eGFP-Cre" in the label of Figure 4I.

(3) While this is intrinsically an issue with the Foxd1-cre model, the observation that was made that only 60% of all stromal cells lost Nphp2 (Figure 4J) is concerning to make such a strong conclusion that NPHP2 in the stromal compartment is not important to the NPHP phenotype (fibrosis or cysts).

We have expanded the analysis to P56 and the results are presented in Figure 4—figure supplement 1 and described in the Results section. In brief, *Invs^flox/flox^;Foxd1-Cre* mice displayed normal kidney weight, KBW ratio, kidney function and histology at P56. By comparison, *Invs^flox/flox^;Cdh16-Cre* kidneys were already cystic at P14. We also added "up to the young adult stage" to make our conclusion regarding the lack of renal phenotypes in interstitial KO of *Invs* more precise.

NPHP2 and Cilia: (1) While the finding that the CDCA may be critical to NPHP, as well as ADPKD, is novel, the provided data is only descriptive. It would improve the paper drastically if the authors could provide data suggesting whether NPHP2 functions in the same pathway as PC1 or PC2 or a different one.

We agree with the reviewer whether INVS functions in the same pathway as polycystins is an interestingly question. However, we feel it is out of the scope of this manuscript and would pursue this research direction in our future studies.

(2) For their cell analyses, the authors should show successful loss of NPHP2 in their CRISPR clones as well as also analyze PC1.

We added Figure 6—figure supplement 1 to provide more detailed characterization of the CRISPR clones.

We did not analyze PC1 for the lack of a reliable antibody. We also feel it outside of the scope of this manuscript.

(3) Fibrosis was not analyzed.

We have now analyzed markers of myofibroblast activation and fibrosis via immuno-staining and Western Blot at P21 and detected a reduction of these markers in double mutants. The new results were added to Figure 5 and described in Results.

HDAC preclinical trial: This data is not connected to anything else within the manuscript. It utilized a different cre than for all the other experiments and is not mechanistically linked to the story of whether NPHP2 function in epithelial or stromal cells is important to pathogenesis. Unless the authors can link the stories better or highlight why the HDAC function is important to the observations, this reviewer would suggest removing Figure 7.

Given the current lack of treatment for NPHP, we feel it important to communicate the results to the research community even though the molecular mechanism remains to be defined.